# YOSO: You-Only-Sample-Once via Compressed Sensing for Graph Neural Network Training

## Abstract

Graph Neural Networks (GNNs) have become essential tools for analyzing structured data across various domains. In GNNs, sampling is critical for reducing training latency by limiting the number of nodes processed during training, especially for large-scale applications. However, as the demand for better prediction performance increases, existing sampling algorithms become more complex, introducing significant overhead in the training process. To address this issue, we introduce YOSO (You-Only-Sample-Once), an algorithm designed to achieve highly efficient training while preserving prediction accuracy in downstream tasks. YOSO proposes a compressed sensing-based sampling and reconstruction framework, where nodes are sampled once at the input layer, followed by a lossless reconstruction at the output layer during each epoch. This approach not only avoids costly computations, such as orthonormal basis, but also guarantees high-probability accuracy retention, equivalent to full node participation. Experimental results on both node classification and link prediction tasks demonstrate the effectiveness and efficiency of YOSO, reducing GNN training by an average of around 75% compared to state-of-the-art methods, while maintaining accuracy on par with top-performing baselines.

## 1 Introduction

Graph Neural Networks (GNNs) (Kipf & Welling, 2016; Hamilton et al., 2017; Veličković et al., 2017; Chen et al., 2018; Chiang et al., 2019; Zou et al., 2019) have become pivotal in modeling structured data across various domains, such as social network analysis (Guo & Wang, 2020), protein interactions (Réau et al., 2023), and transportation systems (Liu et al., 2021a). As graphs rapidly grow, long training time in GNNs becomes a crucial factor impeding the wide utilization of GNNs. To mitigate the issue, various sampling strategies such as node-wise (Hamilton et al., 2017; Chen et al., 2017), layer-wise (Chen et al., 2018; Zou et al., 2019; Huang et al., 2018), and subgraph-based methods (Chiang et al., 2019; Zeng et al., 2019) have been developed. These techniques facilitate mini-batch training, which reduces the amount of memory required to sustain the training process and potentially speeds up convergence. However, with the increasing complexity of sampling algorithms, GNNs have struggled to maintain training efficiency in large-scale applications (Gong et al., 2023) and large graph datasets (e.g., OGB (Hu et al., 2020) and IGB (Khatua et al., 2023)).

Theoretically, the challenge in sampling stems from the biases and variances introduced when altering the data distribution during training (Huang et al., 2018). Unlike the inherently unbiased and variance-free GCN (Kipf & Welling, 2016; Huang et al., 2018) that utilize all training nodes, easily computable sampling methods struggle to accurately estimate both graph structure and features or embeddings (Jin et al., 2020), potentially degrading the learning outcomes. To achieve more precise estimates, such as ensuring unbiasedness and variance reduction, subsequent methods have become increasingly complex, focusing predominantly on reducing variance to improve accuracy but often at the expense of increased computational demands. However, this contradicts the initial goal of sampling–to reduce computational load–and highlights a significant gap in current research: finding a method that achieves both high accuracy and efficiency. This gap is represented by the difference between the target goal (i.e., the seven-pointed red star) and other sampling schemes in Figure 1(a).

To reveal the large overhead introduced by sampling in GNN training, we conduct empirical evaluations for state-of-the-art (SOTA) sampling schemes with Reddit dataset (Hamilton et al., 2017)(de-

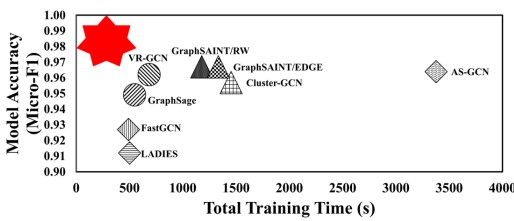 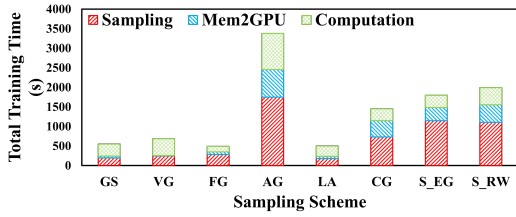

(a) Total training time v.s. model accuracy  (b) Total training time breakdown on Reddit

Figure 1: Total training time (with breakdown) and model accuracy for different sampling schemes, including GS (GraphSage (Hamilton et al., 2017)), VG (VR-GCN (Chen et al., 2017)), FG (Fast-GCN (Chen et al., 2018)), AG (AS-GCN (Huang et al., 2018)), LA (LADIES (Zou et al., 2019)), CG (Cluster-GCN (Chiang et al., 2019)) and two versions of GraphSAINT (Zeng et al., 2019): S_EG (EDGE) and S_RW (Random Walk), on Reddit dataset (Hamilton et al., 2017). The seven-pointed red star marks the contribution of this paper.

tailed setup and environments consistent with Section 6.1). As shown in Figure 1(b), we break down the total training time into three non-overlapping components: (1) Sampling, (2) Mem2GPU: refers to transferring mini-batches from memory to GPU memory, and (3) Computation: all processes on GPU–forward/backward propagation and parameter updates etc. Our results indicate that sampling can account for 35.7% to 64% of the total training time across various sampling algorithms, making it a significant overhead when considering both training efficiency and model accuracy. For instance, as a layer-wise sampling method, AS-GCN (Huang et al., 2018) spends 55.6% of the total training time (i.e., 3376.5 seconds) on sampling but only achieves suboptimal accuracy (0.964, which is 0.03 below the best-performing method, as shown in Figure 1(a)). In contrast, subgraph-based sampling methods, although achieving the highest model accuracy at 0.967, incur the most significant overhead, with sampling accounting for up to 64% of the total training time. Node-wise sampling methods fall between these two paradigms in terms of overall performance. For example, VR-GCN spends 685.72 seconds on sampling and achieves a Micro-F1 score of 0.962.

To address these inefficiencies and the identified research gap, we propose a novel approach, YOSO (You-Only-Sample-Once), which innovatively applies compressed sensing (Candes & Tao, 2006) (CS) to GNN sampling. YOSO reimagines the feature or embedding matrix as multi-channel signals, utilizing adapted compressed sensing techniques to reduce the amount of computation involved in the training by transferring the feature matrix to another domain with high sparsity. This approach enables training with only a fraction $M$ of nodes from a graph with $N$ nodes (where $M \ll N$), and reconstructs the training effect as if all $N$ nodes were used. The nearly lossless reconstruction feature of YOSO guarantees that model performance closely aligns with zero bias and variance. Moreover, the sampling process in YOSO is designed to occur only once at the beginning of the training. This involves determining the sampling set and sampling matrix based on the specific characteristics of the graph. Subsequently, the reconstruction process takes place after each forward propagation. This is done by utilizing the loss generated from the reconstructed embedding matrix to guide the backward propagation. This innovative approach not only streamlines the entire training process by eliminating the need for continuous resampling but also ensures that every step of learning is informed by an optimally reconstructed data state, significantly enhancing both the efficiency and efficacy of the model training. We summarize our contributions below.

- We proposed a novel approach named YOSO that significantly reduces GNN training time while maintaining strong prediction accuracy across various downstream tasks by performing only-once sampling for the entire training process.
- YOSO eliminates the need for expensive computations typically associated with combining compressed sensing with GNN sampling (e.g., determining the orthonormal basis and sampling matrix), thereby making the sampling process highly efficient.
- Experimental results demonstrate the effectiveness of YOSO on both node classification and link prediction tasks. Specifically, YOSO significantly reduces overall training time by an average of around 75% while preserving model accuracy. Ablation studies further reveal that YOSO achieves near-zero bias and variance, effectively reconstructing the embedding matrix with minimal error.

## 2 RELATED WORK

### 2.1 SCHEMES FOR LARGE-SCALE GNN TRAINING

To address the efficiency issue of large-scale GNN training, different schemes have been proposed at the algorithmic level (Zhang et al., 2023), such as Historical Embedding (Chen et al., 2017; Fey et al., 2021), Linearization (Frasca et al., 2020; Abu-El-Haija et al., 2021), Graph Condensation & Distillation (Zheng et al., 2024; Wu et al., 2022), and sampling-based methods. For the first three schemes, please refer to Appendix A for a detailed description, where we discuss their differences and connections with sampling-based methods. The scope of this paper focuses on sampling-based methods.

A widely accepted criterion (Liu et al., 2021b) divides current different sampling methods into three categories: node-wise sampling, layer-wise sampling, and subgraph-based sampling, depending on the granularity of the sampling operation during mini-batch generation.

**Node-wise Sampling:** This fundamental approach, pioneered by works such as GraphSage (Hamilton et al., 2017) and others (Ying et al., 2018; Chen et al., 2017; Dai et al., 2018), involves sampling at the individual node level. Each node's neighbors are selected according to specific probabilities. For example, GraphSage samples $k-$hop neighbors at varying depths with the sampling sizes, for each depth tailored to optimize model performance. This approach, while simple and effective, has been criticized for its exponential increase in sampling time complexity as the number of GNN layers grows.

**Layer-wise Sampling:** Developed to address the exponential growth in computational complexity as GNNs depth increases in node-wise sampling, this method samples multiple nodes simultaneously in one step. Techniques like FastGCN (Chen et al., 2018) reframe GNN loss functions as integral transformations and utilize importance sampling and Monte-Carlo approximation to manage variance. Further developments, such as AS-GCN (Huang et al., 2018) and LADIES (Zou et al., 2019), focus on maintaining sparse connections between sampled nodes to aid convergence. However, these methods tend to introduce additional complexity and computational cost.

**Subgraph-based Sampling:** This approach forms mini-batch training subgraphs using expensive graph partitioning algorithms. Cluster-GCN (Chiang et al., 2019) partitions the full graph into clusters, sampling these clusters to create subgraphs for training batches. GraphSAINT (Zeng et al., 2019) dynamically estimates sampling probabilities for nodes and edges to form subgraphs over which the full GNN model is trained. While these techniques typically improve model accuracy, they also lead to longer training time.

### 2.2 COMPRESSED SENSING

CS (Candes & Tao, 2005; 2006) is a framework that enables perfect recovery of data from a significant small number of measurements. Assuming a signal $\mathbf{x} \in \mathcal{R}^N$ can be sparsely represented as $\mathbf{x} = \mathbf{U}\hat{\mathbf{x}}$, where $\hat{\mathbf{x}} \in \mathcal{R}^K$ is sparse, i.e., $\|\hat{\mathbf{x}}\|_0 \ll K$. $\mathbf{U} \in \mathcal{R}^{N \times K}$ usually is a known transformation, the measurement process is modeled as $\mathbf{y} = \mathbf{\Phi}\mathbf{x} = \mathbf{\Phi}\mathbf{U}\hat{\mathbf{x}}$, with $\mathbf{\Phi} \in \mathbb{R}^{M \times N}$ and $M \ll N$. $\mathbf{\Phi}$ is called sensing matrix (Candes & Tao, 2006) or sampling matrix (Anis et al., 2016). Subsequently, CS uses $\mathbf{y} \in \mathcal{R}^M$ instead of the original data $\mathbf{x} \in \mathcal{R}^N$ for processes, such as computation (Shi et al., 2019) or network transmission (Haupt et al., 2008). Since $M \ll N$, the cost of using $\mathbf{y}$ is significantly lower than directly using $\mathbf{x}$. After the process step is completed, the original data $\mathbf{x}$ needs to be reconstructed from $\mathbf{y}$. The Restricted Isometry Property (RIP) provides a necessary condition for successfully recovering $\mathbf{x}$ from $\mathbf{y}$, which guarantees that $\mathbf{\Phi}\mathbf{U}$ should preserve:

$$(1 - \delta_k)\|\hat{\mathbf{x}}\|_2^2 \le \|\mathbf{\Phi}\mathbf{U}\hat{\mathbf{x}}\|_2^2 \le (1 + \delta_k)\|\hat{\mathbf{x}}\|_2^2,$$

where $\delta_k \in (0, 1)$ is a constant. If the chosen $\mathbf{\Phi}$ and $\mathbf{U}$ satisfy the RIP, then $\hat{\mathbf{x}}$ can be reconstructed perfectly from $\mathbf{y}$ by solving an $\ell_1$-minimization problem

$$\mathrm{argmin}_{\hat{\mathbf{z}} \in \mathcal{R}^K} \|\hat{\mathbf{z}}\|_1 \quad s.t. \quad \mathbf{y} = \mathbf{\Phi}\mathbf{U}\hat{\mathbf{z}} \tag{1}$$

## 3 PRELIMINARIES

**Graph Neural Networks.** GNNs operate on graphs $G = \{V, E, \hat{\mathbf{A}}, \mathbf{X}\}$, where $V = \{1, 2, \ldots, N\}$ represents the set of nodes, $E = \{(i, j) \mid i, j \in V\}$ defines the edges, and $\hat{\mathbf{A}} \in \mathbb{R}^{N \times N}$ is a matrix

that encoding the connections between nodes, i.e., adjacency matrix or normalized Laplacian matrix. The initial node features are stored in the matrix $\mathbf{X} \in \mathbb{R}^{N \times d}$, where $d$ is the feature dimension. GNNs iteratively learn node embeddings $\mathbf{H}^{(l)}$ through a layer-specific transformations governed by parameters $\theta^{(l)}$, expressed as $\mathbf{H}^{(l)} = f_{\theta^{(l)}}(\mathbf{H}^{(l-1)}, \hat{\mathbf{A}})$, $l = 1, 2, ..., L$, where $L$ represents the number of layers, with $\mathbf{H}^{(0)} = \mathbf{X}$.

**Sampling-based GNNs.** To manage computational and storage complexity, a class of GNNs employs sampling techniques, where a subset $V' \subset V$ of nodes is selected based on certain rules $\mathcal{P}$, such as importance sampling and Monte Carlo estimation (Chen et al., 2018). The node embeddings $\mathbf{H}^{(l)}$ are then approximated as $f_{\theta^{(l)}}(\mathbf{H}^{(l-1)}_{[V']}, \hat{\mathbf{A}})$, where $[V']$ denotes the indices corresponding to $V'$, reducing the data need to be processed. However, many sampling methods have increasingly complicated the computation of $\mathcal{P}$ to achieve more accurate approximations, leading to a growing overhead in sampling time.

**Apply CS to GNNs.** Assume there exists a matrix $\mathbf{U}^{(l)} \in \mathcal{R}^{M \times N}$ such that: $\mathbf{H}^{(l)} = \mathbf{U}^{(l)}\hat{\mathbf{H}}^{(l)}$, $l = 1, 2, ..., L$, where $\hat{\mathbf{H}}^{(l)}$ is sparse, i.e., $\hat{\mathbf{H}}^{(l)}$ contains at most $K$ non-zero rows, noted as $\|\hat{\mathbf{H}}^{(l)}\|_{0,row} \leq K$. The set of indices corresponding to the non-zero rows in $\hat{\mathbf{H}}^{(l)}$ is called the support. The existence of such $\mathbf{U}^{(l)}$ is a necessary condition for CS to operate on GNNs. Fortunately, $\mathbf{U}^{(l)}$ that satisfying $\mathbf{H}^{(l)} = \mathbf{U}^{(l)}\hat{\mathbf{H}}^{(l)}$ where $\|\hat{\mathbf{H}}^{(l)}\|_{0,row} \leq k$, exists. The existence has been proven in studies of graph signal processing (Isufi et al., 2024; Bo et al., 2023; Tsitsvero et al., 2016; Puy et al., 2018; Chen et al., 2015). $\mathbf{U}^{(l)}$ can be derived from the properties of graph structure, i.e., normalized Laplacian matrix, and possesses orthogonality: $\mathbf{U}^{(l)}[\mathbf{U}^{(l)}]^{\mathrm{T}} = [\mathbf{U}^{(l)}]^{\mathrm{T}}\mathbf{U}^{(l)} = \mathbf{I}$. Let $\mathbf{T}^{(l)} \in \mathbb{R}^{M \times d}$ where $M \ll N$, be the measurements, computed as:

$$\mathbf{T}^{(l)} = \mathbf{\Phi}^{(l)}\mathbf{U}^{(l)}\hat{\mathbf{H}}^{(l)} \tag{2}$$

where $\mathbf{\Phi}^{(l)} \in \mathcal{R}^{M \times N}$ is the sampling matrix. To reconstruct the original sparse $\hat{\mathbf{H}}^{(l)}$, the following optimization problem need to be solved:

$$\mathrm{argmin}_{\tilde{\mathbf{H}}^{(l)}}\|\tilde{\mathbf{H}}^{(l)}\|_{2,1} \quad \text{s.t.} \quad \mathbf{T}^{(l)} = \mathbf{\Phi}^{(l)}\mathbf{U}^{(l)}\tilde{\mathbf{H}}^{(l)} \tag{3}$$

where $\|\cdot\|_{2,1}$ is $l_{2,1}$ norm (Liu et al., 2018). Perfect reconstruction requires that the matrix $\mathbf{\Phi}^{(l)}\mathbf{U}^{(l)}$ satisfies RIP:

$$(1 - \delta_k)\|\hat{\mathbf{H}}^{(l)}\|_F^2 \leq \|\mathbf{\Phi}^{(l)}\mathbf{U}^{(l)}\hat{\mathbf{H}}^{(l)}\|_F^2 \leq (1 + \delta_k)\|\hat{\mathbf{H}}^{(l)}\|_F^2 \tag{4}$$

where $0 < \delta_k < 1$, and $\|\cdot\|_F$ is the Frobenius norm. After obtaining $\tilde{\mathbf{H}}^{(l)}$ through Equation (3), the original $\mathbf{H}^{(l)}$ can be reconstructed as:

$$\mathbf{H}^{(l)} = [\mathbf{U}^{(l)}]^{\mathrm{T}}\tilde{\mathbf{H}}^{(l)} \tag{5}$$

## 4 DISCUSSION OF COMPRESSED SENSING AS SAMPLING FOR GNNS

**Reason why CS can be used as sampling.** As discussed in Section 3, the goal of sampling is to select a $V'$ that $V' \subset V$ and perform GNN computations on it. Meanwhile, $\mathbf{T}^{(l)} \in \mathbb{R}^{M \times d}$, where the rows of $\mathbf{T}$ form a subset of $V$. Therefore, using $\mathbf{T}$ for GNN computation achieves the same effect as traditional sampling.

**Benefits of compressed sensing as sampling.** Assuming a $\mathbf{\Phi}^{(l)}$ that satisfies all requirements exists (the existence proof and specific form are provided in Section 5.3), CS offers two main advantages over other sampling methods. Firstly, input matrix $\mathbf{H}^{(0)} = \mathbf{X} \in \mathbb{R}^{N \times d}$ can be sampled into a much smaller $\mathbf{T}^{(0)} \in \mathbb{R}^{M \times d}$, significantly reducing computational and storage requirements while retaining essential information. Second, CS enables lossless reconstruction at the output layer, allowing $\mathbf{T}^{(L)} \in \mathcal{R}^{M \times d}$ accurately expanded back to $\mathbf{H}^{(L)} \in \mathcal{R}^{N \times d}$, as if all nodes were involved in the whole computation. Thus, a smaller sample size can effectively emulate the full training set, achieving high accuracy and reduced sampling time.

We can obtain lossless $\mathbf{H}^{(l)} \in \mathbb{R}^{N \times d}$ from $\mathbf{T}^{(l-1)} \in \mathbb{R}^{M \times d}$. This lossless property ensures that the model retains all information, thereby enhancing accuracy. Specifically:

$$\mathbf{H}^{(l)} = f_{\theta^{(l)}}\left(Rec\left\{\mathbf{T}^{(l-1)}\right\}, \hat{\mathbf{A}}\right) \tag{6}$$

where $Rec\{\cdot\}$ represents the processing of reconstruction, i.e., solving the optimization problem in Equation (3) and Equation (5). However, the iterative processes outlined in Equation (6) is highly inefficient and has the following challenges:

- **Expensive Computations of Orthonormal Basis $\mathbf{U}^{(l)}$ and Sampling Matrix $\mathbf{\Phi}^{(l)}$.** Determining appropriate orthonormal bases $\mathbf{U}^{(l)}$ and sampling matrices $\mathbf{\Phi}^{(l)}$ for $l = 1, ..., L$ is time-consuming. While Section 3 theoretically confirms the existence of $\mathbf{U}^{(l)}$, its practical computation is costly, often requiring matrix decompositions with an average time complexity of $O(n^3)$. Since $\mathbf{H}^{(l)}$ changes across GNN layers, a single $\mathbf{U}^{(l)}$ is unlikely to fulfill the sparsity requirements for all layers, necessitating $(L+1)$ separate decompositions. Similarly, $\mathbf{\Phi}^{(l)}$ must adapt to changes in $\mathbf{U}^{(l)}$ to maintain RIP, requiring an additional $(L+1)$ adjustments. Thus, determining both $\mathbf{U}^{(l)}$ and $\mathbf{\Phi}^{(l)}$ involves a total of $2(L+1)$ costly computations during training.
- **Accurate but Time-inefficient Reconstruction.** The original approach reconstructs $\mathbf{H}^{(l)}$ at every layer before proceeding to the next layer's computation to minimize error propagation, as described in Equation (6). However, this incurs significant computational overhead. The fastest known reconstruction algorithm has an average time complexity of $O(nm)$ (Maleki, 2010), where $n$ is the signal dimension and $m$ is the measurement length. For GNNs, this translates to an average reconstruction time complexity of $O(dM)$ per layer, resulting in a total cost of $O(dML)$ for an $L$-layer GNN. Such overhead greatly reduces training efficiency.

Consequently, to effectively integrate CS into GNNs and ensure its efficiency, two obstacles must be overcame in YOSO design:

**Obstacle I. Working with Unknown $\mathbf{U}$ and Universal $\mathbf{\Phi}$.** Given the high computational cost of determining $\mathbf{U}^{(l)}$, we need to satisfy or approximate CS's necessary and sufficient condition without explicitly knowing $\mathbf{U}^{(l)}$. Without $\mathbf{U}^{(l)}$, identifying the support and determining essential nodes for reconstruction becomes challenging, complicating the construction of $\mathbf{\Phi}^{(l)}$. Since $\mathbf{\Phi}^{(l)}$ is layer-specific, calculating it for each layer is impractical. Thus, we require a method that works with an unknown $\mathbf{U}$ using a universal sampling matrix $\mathbf{\Phi}$, ensuring $\mathbf{\Phi}$ remains adaptable to any $\mathbf{U}$ while satisfying compressed sensing conditions.

**Obstacle II. Balancing computational efficiency with the need for accurate reconstruction.** If we sample once at the input layer and use these results throughout the GNN computation, followed by reconstruction only at the output layer, this approach requires just one sampling and reconstruction step for the entire training process. Although it may introduce some accuracy loss due to reduced intermediate layer information, it remains efficient if this loss is controllable with a known upper bound, allowing a balance between computational efficiency and model accuracy.

We address the Obstacle I in Section 5.2 (Unknown $\mathbf{U}$) and Section 5.3 (Univerisal $\mathbf{\Phi}$), respectively, and explain how YOSO solve the Obstacle II in Section 5.2.

## 5 METHODOLOGY

In Section 4, we explained why CS can be applied to GNN sampling and highlighted the benefits of this approach compared to other sampling methods. In this section, we provide a detailed description of the YOSO design, including how it addresses Obstacles I and II from Section 4.

### 5.1 OVERALL PROCESS OF YOSO

YOSO is primarily divided into four parts:

(a) **Construction of Sampling Matrix $\mathbf{\Phi}$.** As discussed in Section 4, we require an universal sampling matrix $\mathbf{\Phi}$ to perform one-time sampling and enable the subsequent training process. Therefore, before the training process in YOSO (as indicated in Equation (2)), it is necessary to construct $\mathbf{\Phi}$ first. For a detailed construction of the sampling matrix $\mathbf{\Phi}$, please refer to Section 5.3.

(b) **One-time Sampling.** This process (Equation (2) where $l = 0$) takes node feature $\mathbf{X} = \mathbf{H}^{(0)} \in \mathcal{R}^{N \times d}$ as input and produces $\mathbf{T}^{(0)} \in \mathcal{R}^{M \times d}$, where $M \ll N$. The computation of the orthonormal basis $\mathbf{U}$ is provided in Section 5.2.

(c) **Forward Propagation.** After obtaining $\mathbf{T}^{(0)}$ from one-time sampling, $\mathbf{T}^{(0)}$ is used as the input for the forward propagation process (Equation (7) in Section 5.2). Upon reaching the output of

the $L$-th layer (i.e., the output layer), noted as $\mathbf{Z} \in \mathcal{R}^{M \times d}$, reconstruction is then performed to obtain $\mathbf{H}^{(L)} \in \mathcal{R}^{N \times d}$. Unlike the traditional CS process (Candes & Tao, 2005), where $\mathbf{U}$ is known and the reconstruction process can be directly computed, in the GNN context, $\mathbf{U}$ is unknown. To overcome this, YOSO constructs a joint loss function to determine the optimal $\mathbf{U}$ together with GNN model parameters through the backpropagation process.

(d) **Joint Optimization and Backward Propagation.** YOSO constructs a joint optimization problem to form the loss function. This loss function consists of two parts. The first part is related to reconstruction (Equation (8) in Section 5.2). The unknown $\mathbf{U}$ affects the reconstruction performance (Equation (3) and Equation (5)) and must ensure that the matrix $\hat{\mathbf{H}}^{(L)}$ to be reconstructed remains sparse; The second part is the GNN's inherent loss, which is specific to the learning task. For example, in node classification, a possible loss function is the cross-entropy loss. By combining these two parts, we aim to simultaneously minimize the reconstruction error and the GNN's inherent loss. This results in a joint loss function (Equation (9) in Section 5.2), which is used to update the relevant parameters through backpropagation.

## 5.2 YOSO DESIGN

This section provides a detailed elaboration on the YOSO design.

Unlike the standard GNN training process, YOSO operates within a specific sparse domain (i.e., $\mathbf{U}$) instead of the original data domain. As shown in Algorithm 1, the YOSO training process consists of four key stages: one-time sampling, forward propagation, loss computation, and backward propagation. The process begins with transforming the data into the sparse domain (Line 3), where one-time sampling is performed using the sampling matrix $\mathbf{\Phi}$. The subsequent steps: forward propagation (Lines 4-7), loss computation (Lines 8-10), and backward propagation (Lines 11-17) are carried out entirely within this sparse domain. A detailed explanation of these steps is provided below:

**Initialization process (Line 1).** The parameters are initialized randomly using the Xavier initialization method (Glorot & Bengio, 2010). It is worth noting that $\mathbf{U}$ is not initialized as an orthogonal matrix but is iteratively adjusted to become orthogonal during the training process (Line 16).

**One-time sampling (Line 3).** Given a graph $G = \{V, E, \hat{\mathbf{A}}, \mathbf{X}\}$, where specific $\hat{\mathbf{A}}$ is the normalized Laplacian matrix. We perform the sampling stage only once using the sampling matrix $\mathbf{\Phi}$ on the sparsity domain $\hat{\mathbf{X}}$ as $\mathbf{\Phi U} \hat{\mathbf{X}}$, resulting in $\mathbf{T}^{(0)} \in \mathbb{R}^{M \times d}$, where $M \ll |V| = N$. This process involves the construction of the sampling matrix $\mathbf{\Phi}$, for details, please refer to Section 5.3.

**Forward propagation (Lines 4-7).** The forward propagation of YOSO can be expressed as:

$$\begin{cases} \mathbf{T}^{(l)} = \sigma\left(\mathbf{\Phi}\hat{\mathbf{A}}\mathbf{\Phi}^{\mathrm{T}}\mathbf{T}^{(l-1)}\mathbf{W}^{(l)}\right) & 1 \le l \le L-1 \\ \mathbf{U}, \hat{\mathbf{H}}^{(L)} = Rec\left\{\sigma\left(\mathbf{\Phi}\hat{\mathbf{A}}\mathbf{\Phi}^{\mathrm{T}}\mathbf{T}^{(L-1)}\mathbf{W}^{(L)}\right)\right\} & l = L \end{cases} \tag{7}$$

where $\sigma(\cdot)$ is the activation function, $\mathbf{W}^{(l)}, l = 1, ..., L$ is the $l-$th layer's trainable parameters, $\mathbf{U}$ is the unknown orthonormal basis and the method for addressing this (working with unknown $\mathbf{U}$) will be discussed in the following.

**Loss function and working with unknown $\mathbf{U}$ (Lines 8-10).** We discuss the construction of YOSO's loss function in the following two points (i.e., P1 and P2). These two points mainly explain how this construction effectively overcomes the challenge of working with the unknown $\mathbf{U}$ (Obstacle I from Section 4).

**(P1) Loss function.** The $Rec\{\cdot\}$ in Equation (7) is equal to solve the following optimization problem:

$$\arg\min_{\hat{\mathbf{H}}^{(L)}, \mathbf{U}} \left\{\frac{1}{2}\left\|\mathbf{Z} - \mathbf{\Phi U}\hat{\mathbf{H}}^{(L)}\right\|_F^2 + \lambda\left\|\hat{\mathbf{H}}^{(L)}\right\|_{2,1}\right\} \quad \text{s.t. } \mathbf{UU}^{\mathrm{T}} = \mathbf{U}^{\mathrm{T}}\mathbf{U} = \mathbf{I} \tag{8}$$

where $\mathbf{Z} = \sigma(\mathbf{\Phi}\hat{\mathbf{A}}\mathbf{\Phi}^{\mathrm{T}}\mathbf{T}^{(L-1)}\mathbf{W}^{(L)})$ represents the measurement matrix at the output layer, and $\lambda$ is a hyperparameter controlling the balance between fidelity and sparsity. Equation (8) is a non-trivial optimization problem involving both $\hat{\mathbf{H}}^{(L)}$ and $\mathbf{U}$ due to non-convexity introduced by orthogonality constraint ($\mathbf{UU}^{\mathrm{T}} = \mathbf{U}^{\mathrm{T}}\mathbf{U} = \mathbf{I}$) and the interaction between variables. To overcome it, we perform joint optimization of Equation (8) with the GNN's specific loss function (e.g., cross-entropy in node classification learning task). Let the GNN's loss function be $\mathcal{L}_{GNN}^{\Theta}$, where $\Theta = \{\mathbf{W}^{(1)}, ..., \mathbf{W}^{(L)}\}$

represents the set of all trainable parameters. The joint optimization objective function is defined as:

$$\underset{\hat{\mathbf{H}}^{(L)}, \mathbf{U}, \Theta}{\arg\min} \left\{ \alpha \left( \frac{1}{2} \left\| \mathbf{Z} - \mathbf{\Phi U}\hat{\mathbf{H}}^{(L)} \right\|_F^2 + \lambda \left\| \hat{\mathbf{H}}^{(L)} \right\|_{2,1} \right) + \beta \mathcal{L}_{GNN}^{\Theta}(\overline{\mathbf{H}}^{(L)}) \right\} \quad \text{s.t. } \mathbf{UU}^\mathrm{T} = \mathbf{U}^\mathrm{T}\mathbf{U} = \mathbf{I}$$

(9)

where $\alpha$ and $\beta$ are the hyperparameters to balance the reconstruction loss and GNN loss, and based on the reconstruction process in Equation (5), we have $\overline{\mathbf{H}}^{(L)} \in \mathcal{R}^{N \times d}$, which contains the representations for all nodes in the entire graph, will be reconstructed from the results of $\mathbf{Z}$ and subsequently used in calculating the GNN loss.

---

**Algorithm 1** Forward and Backward Propagation of YOSO

1: Initialize $\Theta$, $\mathbf{U}$, and $\hat{\mathbf{H}}^{(L)}$
2: **while** not converged **do**
3:      Compute $\mathbf{T}^{(0)} = \mathbf{\Phi U}\hat{\mathbf{X}}$
4:      **for** $l = 1$ to $L - 1$ **do**
5:          Compute $\mathbf{T}^{(l)} = \sigma \left( \mathbf{\Phi}\hat{\mathbf{A}}\mathbf{\Phi}^\mathrm{T}\mathbf{W}^{(l)}\mathbf{T}^{(l-1)} \right)$
6:      **end for**
7:      Compute $\mathbf{Z} = \sigma \left( \mathbf{\Phi}\hat{\mathbf{A}}\mathbf{\Phi}^\mathrm{T}\mathbf{W}^{(L)}\mathbf{T}^{(L-1)} \right)$
8:      Compute reconstruction Loss: $\mathcal{L}_{\mathrm{recon}} = \frac{1}{2}\|\mathbf{Z} - \mathbf{\Phi U}\hat{\mathbf{H}}^{(L)}\|_F^2 + \lambda\|\hat{\mathbf{H}}^{(L)}\|_{2,1}$
9:      Compute GNN Loss: $\mathcal{L}_{GNN}^{\Theta}(\overline{\mathbf{H}}^{(L)})$
10:     Compute Total Loss: $\mathcal{L} = \alpha\mathcal{L}_{\mathrm{recon}} + \beta\mathcal{L}_{GNN}^{\Theta}(\overline{\mathbf{H}}^{(L)})$
11:     Compute gradient w.r.t $\Theta$: $\nabla_{\Theta}\mathcal{L} = \alpha\nabla_{\Theta}\mathcal{L}_{\mathrm{recon}} + \beta\nabla_{\Theta}\mathcal{L}_{GNN}^{\Theta}(\overline{\mathbf{H}}^{(L)})$
12:     Compute gradient w.r.t $\mathbf{U}$: $\nabla_{\mathbf{U}}\mathcal{L} = \alpha\nabla_{\mathbf{U}}\mathcal{L}_{\mathrm{recon}} + \beta\nabla_{\mathbf{U}}\mathcal{L}_{GNN}^{\Theta}(\overline{\mathbf{H}}^{(L)})$
13:     Compute gradient w.r.t $\hat{\mathbf{H}}^{(L)}$: $\nabla_{\hat{\mathbf{H}}^{(L)}}\mathcal{L} = \eta_{\hat{\mathbf{H}}^{(L)}}\nabla_{\hat{\mathbf{H}}^{(L)}}\mathcal{L}_{\mathrm{recon}}$
14:     Update $\Theta$: $\Theta \leftarrow \Theta - \eta_{\Theta}\nabla_{\Theta}\mathcal{L}$
15:     Update $\mathbf{U}$: $\mathbf{U}_{\mathrm{temp}} = \mathbf{U} - \eta_{\mathbf{U}}\nabla_{\mathbf{U}}\mathcal{L}$
16:     Project $\mathbf{U}$ onto the Stiefel manifold (Koochakzadeh et al., 2016) to ensure $\mathbf{U}^\mathrm{T}\mathbf{U} = I$
17:     Update $\hat{\mathbf{H}}^{(L)}$: $\hat{\mathbf{H}}^{(L)} \leftarrow \hat{\mathbf{H}}^{(L)} - \eta_{\hat{\mathbf{H}}}\nabla_{\hat{\mathbf{H}}^{(L)}}\mathcal{L}$
18: **end while**

---

**(P2) Working with unknown $\mathbf{U}$.** To tackle the challenge of the unknown $\mathbf{U}$, we treat $\mathbf{U}$ as an optimization target. By leveraging Equation (9), we compute the total loss, which is subsequently used to generate gradients for updating $\mathbf{U}$ through all training processes. A detailed derivation of the gradient of the loss in Equation (9) with respect to $\mathbf{U}$ can be found in Appendix D.1).

**Backward Propagation (Lines 11-17).** The backward propagation process uses the loss generated by Equation (9) to update three parameters, which are $\mathbf{U}$, $\hat{\mathbf{H}}^{(L)}$ and $\Theta = \{\mathbf{W}^{(1)}, ..., \mathbf{W}^{(L)}\}$ through gradient descent. This process results in three gradients, namely $\nabla_{\mathbf{U}}\mathcal{L}$, $\nabla_{\hat{\mathbf{H}}^{(L)}}\mathcal{L}$, and $\nabla_{\Theta}\mathcal{L}$, each corresponding to three learning rates $\eta_{\mathbf{U}}$, $\eta_{\hat{\mathbf{H}}^{(L)}}$, and $\eta_{\Theta}$, respectively. For the detailed setting of hyperparameters used here, i.e., $\alpha$ and $\eta_{\mathbf{U}}$, please refer to Appendix C.4 and the detailed gradient computation list in Appendix D.1.

Through Algorithm 1, we obtain both $\mathbf{U}$ and $\hat{\mathbf{H}}^{(L)}$. With $\mathbf{U}$ now determined, Equation (5) can be used to reconstruct $\mathbf{H}^{(L)}$, which is then applicable to downstream tasks such as link prediction. Compared to Equation (6), the process described in Algorithm 1 achieves improved efficiency at the cost of a slight reduction in accuracy and this loss in accuracy is bounded. For detailed statements and proofs, see Appendix D.4.

## 5.3 Construction of Sampling Matrix $\mathbf{\Phi}$

When the orthonormal basis $\mathbf{U}$ remains unspecified prior to training, a key challenge arises in computing $\mathbf{T}^{(0)} = \mathbf{\Phi U}\hat{\mathbf{X}}$ in Equation (7), as the lack of knowledge about $\mathbf{U}$ complicates the design of $\mathbf{\Phi}$. In traditional CS, $\mathbf{U}$ maps data into a sparse domain, where the support (i.e., the indices of non-zero rows) is explicitly identifiable, and these non-zero rows contain the crucial information. This clarity allows $\mathbf{\Phi}$ to be tailored to the supports. However, without prior knowledge of $\mathbf{U}$, designing a

$\boldsymbol{\Phi}$ that effectively captures the essential information becomes significantly more difficult. Thus, the core challenge lies in designing an robust and universal sampling matrix $\boldsymbol{\Phi}$ that accurately captures the key characteristics of the graph data while remaining compatible with any $\mathbf{U}$ without violating the RIP.

To address this challenge, we propose an approach that integrates the design of a matrix $\hat{\mathbf{S}} \in \mathbb{R}^{M \times N}$, derived from the graph structure, with the construction of the sampling matrix $\boldsymbol{\Phi}$, i.e., $\boldsymbol{\Phi} = \hat{\mathbf{S}} \otimes \boldsymbol{\Sigma}$, where $\boldsymbol{\Sigma} \in \mathbb{R}^{M \times N}$ is a random matrix and $\otimes$ is element-wise production. The matrix $\hat{\mathbf{S}}$ is determined once during the preprocessing phase and remains fixed throughout training. Its design is based on graph-structure for two reasons: (1) the graph structure is invariant, and (2) it reflects the importance of certain nodes, which is crucial for the GNN message-passing process. For the sampling matrix $\boldsymbol{\Phi}$, maintaining a row full rank property is essential. Intuitively, $\boldsymbol{\Phi}$ combines node features or embeddings linearly, using weights corresponding to the indices of the support (non-zero rows). If $\boldsymbol{\Phi}$ is row over-ranked, it introduces redundancy, if row under rank, it results in information loss. Thus, ensuring that $\boldsymbol{\Phi}$ is crucial for effectively capturing the necessary information.

**Construction of $\hat{\mathbf{S}}$.** Considering the normalized Laplacian matrix $\hat{\mathbf{A}} = \mathbf{I} - \mathbf{D}^{-1/2}\mathbf{A}\mathbf{D}^{-1/2}$ where $\mathbf{D}$ and $\mathbf{A}$ denote the degree matrix and adjacency matrix, respectively. The $N$ nodes correspond to $N$ eigenvalues from the spectral decomposition of $\hat{\mathbf{A}}$, denoted as $\{\lambda_1, \ldots, \lambda_N\}$ with $\lambda_i \geq 0$ for any $i$. These eigenvalues capture important structural properties of the graph, where larger eigenvalues correspond to more influential nodes. To construct the sampling probability distribution, we define $P(i) = \frac{\lambda_i}{\sum_{j=1}^{N} \lambda_j}$, assigning node $i$ a sampling probability proportional to its eigenvalue relative to the total eigenvalue sum. Using this distribution, we sample $M$ times to form the $M$ rows of $\hat{\mathbf{S}}$. If node $i$ is sampled, the corresponding row in $\hat{\mathbf{S}}$ includes the 1-hop neighbors of node $i$. Assume node $i$ has $N(i)$ neighbors, each neighbor is randomly sampled with a probability of $\frac{1}{N(i)}$. This construction ensures that $\hat{\mathbf{S}}$ will not contain any all-zero rows, due to the self-loop added by the normalized Laplacian. Consequently, the matrix $\boldsymbol{\Phi} = \hat{\mathbf{S}}\boldsymbol{\Sigma}$ is row full rank (detailed proof in Appendix D.2), avoiding any row rank deficiency issues.

**Construction of $\boldsymbol{\Sigma}$.** Randomness has been shown to play a critical role in achieving the RIP (Baraniuk et al., 2008). Therefore, we define $\boldsymbol{\Sigma}$ as a random matrix. Intuitively, in the absence of precise support knowledge, we estimate the support by randomly sampling $M$ nodes based on eigenvalue weights. The matrix $\boldsymbol{\Sigma}$ must represent the contribution level of each node $i$ to the non-zero rows (the support). For instance, if node $k$ contributes to both nodes $i$ and $j$, $\boldsymbol{\Sigma}$ should quantify $k$'s contribution to each. This is crucial for ensuring accurate reconstruction and satisfying the RIP. For any column $j$ in $\hat{\mathbf{S}}$, assume it contains $g(j)$ non-zero elements. The corresponding elements in $\boldsymbol{\Sigma}$ are assigned random values drawn from a Gaussian distribution $N(0, \frac{1}{g(j)})$. This design effectively captures contribution levels, ensuring compliance with the Restricted Isometry Property.(detailed proof in Appendix D.3).

# 6 EXPERIMENTS

In Section 6.2, we evaluate the training time along with model accuracy across two learning tasks: node classification and link prediction. Also, to investigate convergence performance, we assess the convergence of both the baselines and YOSO in Section 6.3. Finally, we conduct an ablation study on the proposed compensations in Section 6.4. Details on the dataset, baselines, experimental hardware and software configuration can be found in Section 6.1 and Appendix C.1.

## 6.1 EXPERIMENTAL SETTINGS

**Datasets.** For the node classification task, we selected Reddit (Hamilton et al., 2017), ogbn-arxiv and ogbn-products (Hu et al., 2020). For the link prediction task, we used ogbl-ppa, and ogbl-citation2 (Hu et al., 2020). For detailed dataset statistics, data splits and metrics, please refer to Appendix C.2.

**Baselines and Implementation.** The baselines used in this paper include node-wise sampling methods (GraphSage (Hamilton et al., 2017) and VR-GCN (Chen et al., 2017)), layer-wise sampling methods (FastGCN (Chen et al., 2018), AS-GCN (Huang et al., 2018) and LADIES (Zou et al., 2019)) and subgraph-based sampling methods (Cluster-GCN (Chiang et al., 2019) and Graph-SAINT (Zeng et al., 2019)). Notably, several baseline models lacked implementations for link pre-

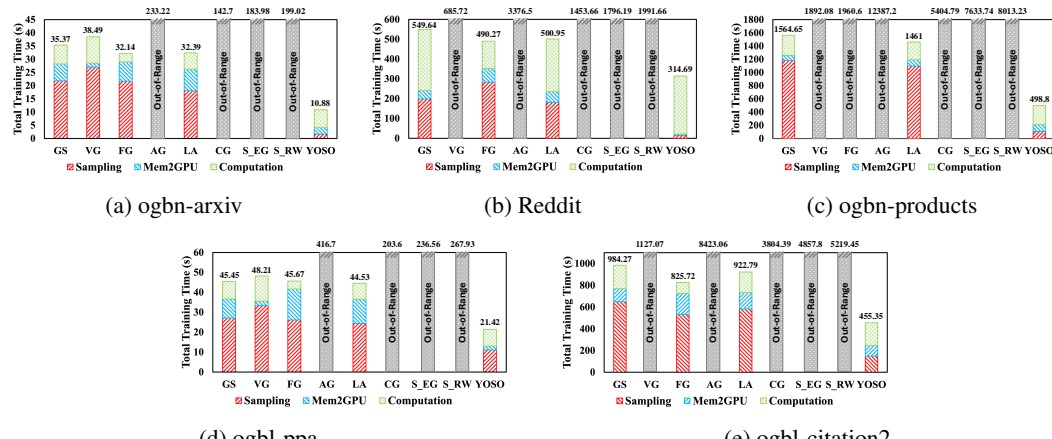

(a) ogbn-arxiv  (b) Reddit  (c) ogbn-products

(d) ogbl-ppa  (e) ogbl-citation2

Figure 2: Total training time comparison with the breakdown times including Sampling, Computation, and Mem2GPU. This evaluation covers two learning tasks across five datasets: (a) to (c) represent the results for the node classification task on ogbn-arxiv (Hu et al., 2020), Reddit (Hamilton et al., 2017), and ogbn-products (Hu et al., 2020), respectively; while (d)-(e) correspond to the link prediction task on ogbl-ppa (Hu et al., 2020) and ogbl-citation2 (Hu et al., 2020). We use the same model name abbreviations as in Figure 1.

diction, prompting us to modify them accordingly. Detailed information on the source code for these baselines, the YOSO implementation, and other related materials can be found in Appendix C.3.

**Hyperparameter Setting.** All experiments are conducted using a two-layer GNN. Detailed hyperparameter settings are described in Appendix C.4.

## 6.2 OVERALL COMPARISON

In this section, we evaluate baselines and YOSO with two key metrics: model accuracy (varies with different datasets and tasks) and total training time. The training time is broken down into three non-overlapped parts: Sampling, Mem2GPU, and Computation.

**Node Classification Task:** First, YOSO achieves the shortest total training time with an average of 75.3% reduction across all datasets compared to all baselines as shown in Figure 2. For example, YOSO reduces around 95% total training time from 233.22 seconds (ogbn-arxiv/AS-GCN) and 12,387.2 seconds (ogbn-products/AS-GCN) to 199.02 and 8,013.23 seconds, respectively. The main reason is that YOSO significantly reduces the sampling time while introducing a little reconstruction overhead. As shown in Figure 2(a)-(c), the most substantial sampling time reduction occurs on the Reddit dataset, where YOSO achieved a 99% decrease, cutting the sampling time from 1149.02 seconds for GraphSAINT-EDGE and 1107.54 seconds for Random Walk to just 15.13 seconds. On average, YOSO reduced sampling time by approximately 95.7% compared to all other baselines.

For model accuracy shown in Table 1, YOSO consistently matches or closely approaches the top performers. For example, YOSO obtains an accuracy of 0.71 on ogbn-arxiv, just 0.01 below GraphSage. On Reddit, it achieves the highest score of 0.967, matching GraphSAINT-Random Walk, and on ogbn-products, it reaches 0.787, slightly trailing GraphSAINT-EDGE's 0.792.

**Link Prediction Task:** For total training time, similar to the node classification task, YOSO achieves the best training time with a 72.13% average training time decrease across all datasets for the link prediction. For example, YOSO decreases the training time for the ogbl-ppa dataset from 44.53 seconds with AG-GCN to 21.42 seconds, and for the ogbl-citation2 dataset from 8423.06 seconds with AG-GCN to 455.35 seconds.This improvement is consistent with the node classification task, where YOSO achieves considerable reductions in sampling time while introducing minimal reconstruction overhead. As depicted in Figure 2(d)-(e), YOSO achieves an average sampling time reduction of about 80.5% across all datasets. As for model accuracy, outlined in Table 1, YOSO

Table 1: Model accuracy results for different sampling schemes on node classification and link prediction tasks. For specific evaluation metrics on each dataset, please refer to Table 5.

| Different Sampling Schemes | Dataset | | | | |
|---|---|---|---|---|---|
| | Node Classification | | | Link Prediction | |
| | ogbn-arxiv | Reddit | ogbn-products | ogbl-ppa | ogbl-citation2 |
| GraphSage | **0.72** | 0.949 | 0.772 | 0.1704 | **0.8054** |
| VR-GCN | 0.697 | 0.962 | 0.699 | 0.1704 | 0.7967 |
| FastGCN | 0.438 | 0.927 | 0.404 | 0.1088 | 0.6555 |
| AS-GCN | 0.687 | 0.964 | 0.51 | 0.1245 | 0.6593 |
| LADIES | 0.649 | 0.927 | 0.501 | 0.1131 | 0.6693 |
| Cluster-GCN | 0.653 | 0.966 | 0.769 | 0.2053 | 0.7904 |
| GraphSAINT-EG | 0.702 | **0.967** | **0.792** | 0.2143 | 0.8039 |
| GraphSAINT-RW | 0.701 | **0.967** | 0.783 | **0.2263** | **0.8054** |
| YOSO | **0.72** | **0.967** | 0.787 | 0.2238 | 0.8025 |

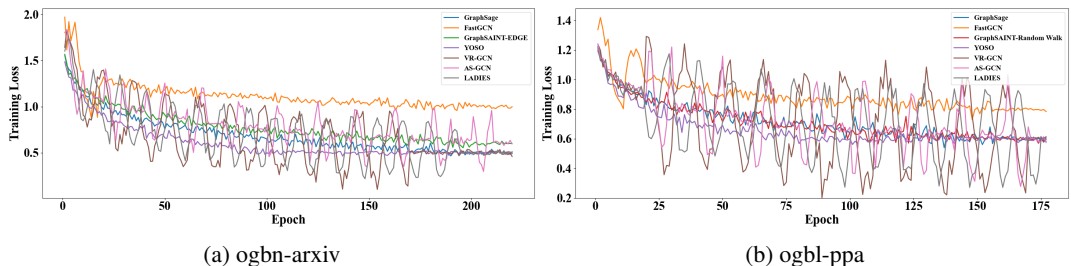

(a) ogbn-arxiv                  (b) ogbl-ppa

Figure 3: Training loss and epoch curves for YOSO and baselines on two benchmark datasets.

maintained results with only a very small gap–0.0025 on ogbn-arxiv and 0.0029 on ogbl-citation2–compared to the best results achieved by GraphSAINT-Random Walk and GraphSage, respectively.

In summary, for both tasks of node classification and link prediction, by combining high accuracy with substantial reductions in sampling and total training time, YOSO demonstrates its efficiency in GNN training and significantly improves both sampling and total training times across all datasets while maintaining competitive accuracy, highlighting its effectiveness compared to the baselines on the node classification task.

## 6.3 CONVERGENCE COMPARISON

We investigate YOSO's convergence performance compared to other baselines. Specifically, we select ogbn-arxiv and ogbl-ppa as representatives for node classification and link prediction, respectively. The training loss-epoch curves are shown in Figure 3.

In both experiments, YOSO consistently outperformed the baselines in terms of convergence speed and stability. On the ogbn-arxiv dataset, YOSO reached a lower training loss more rapidly than GraphSAGE, GraphSAINT-EDGE, and FastGCN, with significantly fewer oscillations, indicating a more stable and efficient training process. Similarly, on the ogbl-ppa dataset, YOSO demonstrated faster convergence and maintained a smoother training loss curve, while the baselines, especially FastGCN, exhibited more fluctuations. These results suggest that YOSO not only accelerates the convergence process but also ensures a more stable training path compared to existing sampling methods, highlighting its effectiveness in GNN training

## 6.4 ABLATION STUDY

In this subsection, we explore how YOSO's total training time and model accuracy vary with different sampling sizes $M$ and evaluate reconstruction effectiveness by comparing the $\mathbf{H}^{(L)}$ matrix generated without sampling to the $\tilde{\mathbf{H}}^{(L)}$ matrix produced by YOSO's sampling-reconstruction process, with the differences visualized with heatmaps.

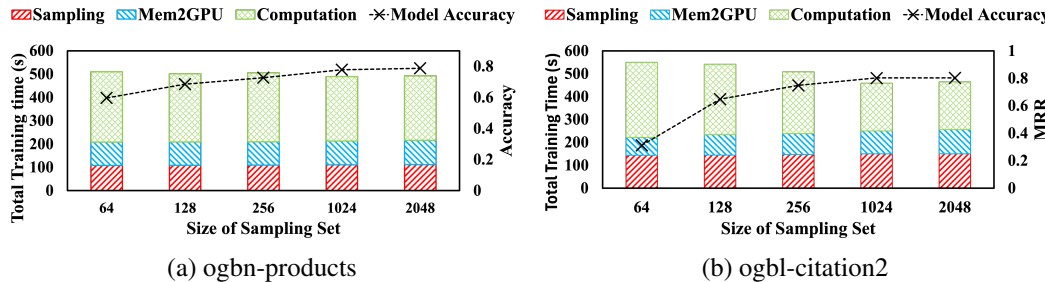

(a) ogbn-products     (b) ogbl-citation2

Figure 4: Total training time (including its breakdown) and model accuracy for YOSO with different sampling sizes: (a) for the node classification learning task on the ogbn-products dataset, and (b) for the link prediction learning task on the ogbl-citation2 dataset.
.

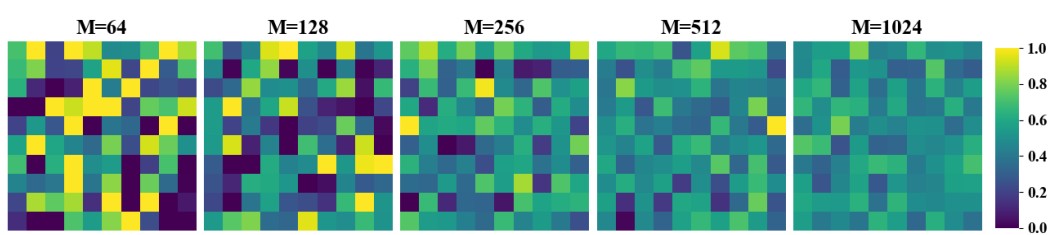

Figure 5: Reconstruction effectiveness visualized via heatmap. Using the ogbn-products dataset, 10 nodes are randomly selected from the training set, and for each node, 10 embedding dimensions are randomly picked. The heatmap shows the absolute differences between original and reconstructed embeddings for these elements. $M$ is the size of the sampling set.

**Varying sampling size $M$:** We examine how total training time (including breakdown) and model accuracy vary with $M$ values, specifically $M = \{64, 128, 256, 1024, 2048\}$, as shown in Figure 4. The results indicate that YOSO's sampling time remains stable across different $M$, ranging from 107.94 to 111.53 seconds on ogbn-products and 143.56 to 149.65 seconds on ogbl-citation2, showing minimal impact from $M$. In contrast, as $M$ decreases, computation time increases, reflecting more iterations needed for convergence (e.g., rising from 275.98s at $M = 2048$ to 301.94s at $M = 64$ on ogbn-products, with a similar trend on ogbl-citation2). Model accuracy improves with larger $M$, eventually stabilizing; it rises from 0.597 to 0.7873 on ogbn-products and from 0.312 to 0.8025 on ogbl-citation2. These findings highlight YOSO's efficient sampling and improved accuracy and convergence with larger $M$.

**Reconstruction effectiveness:** The heatmap in Figure 5 shows the reconstruction effectiveness for different sampling sizes $M$. Each $10 \times 10$ block represents the absolute difference between reconstructed embeddings from our two-layer GNN sampling and those computed with all neighbors (without sampling). As $M$ increases, reconstruction accuracy improves, enhancing overall model accuracy. However, beyond a certain point, such as $M = 512$ in Figure 5, further increases in $M$ offer diminishing returns in both reconstruction quality and model accuracy. This suggests there is an optimal $M$ that balances reconstruction quality and computational efficiency.

## 7 CONCLUSION

In this paper, we introduce YOSO (You Only Sample Once), a novel algorithm aimed at significantly enhancing the efficiency of GNN training without sacrificing prediction accuracy. By leveraging a compressed sensing-based sampling and reconstruction framework, YOSO performs node sampling only once at the input layer, followed by a lossless reconstruction at the output layer during each training epoch. Our experimental results demonstrate that YOSO can achieve up to 75% reduction of existing state-of-the-art methods while achieving accuracy comparable to top-performing baselines.

**Ethics Statement:** In this paper, we present a technique grounded in compressed sensing that addresses the growing computational demands of GNN sampling schemes. Our approach significantly reduces sampling time and overall GNN training duration without compromising model accuracy, thereby enhancing the efficiency of graph neural network training. This improvement holds potential for a wide range of applications, such as recommendation systems and social network analysis, and bioinformatics. We believe that our method contributes positively to the advancement of machine learning research by promoting computational efficiency. Although we do not anticipate any immediate negative ethical implications or societal concerns from our approach, it's important to acknowledge that machine learning technologies, including graph-based methods, have broader impacts. Therefore, responsible implementation is crucial to ensure that such technologies are applied in a manner that promotes fairness and beneficial societal outcomes.

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

# A ADDITIONAL RELATED WORKS AND DISCUSSION

**Graph Condensation&Distillation:** Graph Condensation Gao et al. (2024) and Graph Distillation Tian et al. (2023) are methods designed to enhance computational efficiency. They achieve this by shrinking large-scale graphs into smaller ones while preserving essential structural and feature information. Alternatively, they replace complex GNN models with approximate and computationally simpler models, such as MLPs (Ramchoun et al., 2016). However, these kinds of processes introduce additional computational overhead and may result in the loss of important information, potentially leading to a decrease in model performance. For example, GCond (Jin et al., 2021) leverages a gradient matching framework to condense large graphs into significantly smaller synthetic graphs. It optimizes node features as free parameters and models synthetic graph structures as functions of these features, ensuring that training trajectories on the condensed graph mimic those on the original graph. Another work, GC-SNTK Wang et al. (2024), reformulates graph condensation as a Kernel Ridge Regression (KRR) task, replacing computationally intensive GNN training with a Structure-based Neural Tangent Kernel (SNTK). This approach captures both node feature interactions and structural relationships, enabling efficient graph condensation while maintaining strong generalization across GNN architectures.

**Historical Embedding.** This class of methods is not independent of sampling. Instead, they are often integrated with existing sampling strategies to improve specific aspects of sampling performance, such as estimated variance (Chen et al., 2017), or expressiveness (Fey et al., 2021). For example, VR-GCN (Chen et al., 2017) utilizes historical embeddings within node-wise sampling. GNNAutoScale (Fey et al., 2021) incorporates the concept of historical embeddings within subgraph-based sampling. Although historical embedding can be effective in terms of accuracy, it often comes with high computational complexity.

**Linearization.** This stream of works (Abu-El-Haija et al., 2021; Frasca et al., 2020) aims to simplify the training and inference processes by removing the nonlinear components (e.g., activation functions or deep iterative propagation) inherent in traditional GNN models. This simplification achieves computational efficiency while preserving essential graph structure and feature information through linear transformations, i.e., SIGN Frasca et al. (2020) or precomputations, i.e., iSVD (Abu-El-Haija et al., 2021). Linearization techniques often involve precomputing graph-based transformations (e.g., matrix products or embeddings) and applying efficient optimization methods (e.g., truncated Singular Value Decomposition (SVD) or matrix factorization) to enable scalable training, particularly for large graphs.

# B ADDITIONAL EXPERIMENTS

## B.0.1 YOSO V.S. GRAPH CONDENSATION&DISTILLATION

Both Graph Condensation and Graph Distillation introduce additional computational overhead and may result in the loss of important information, potentially leading to a decrease in model performance.

To better demonstrate the advantages of YOSO over these schemes, we conducted new experiments comparing YOSO with two classic graph condensation schemes, GCond (Jin et al., 2021) and GC-SNTK (Wang et al., 2024), on the ogbn-arxiv dataset. Both GCond and GC-SNTK use a graph reduction rate of 0.25% on the ogbn-arxiv dataset and are paired with GCN (Kipf & Welling, 2016). We evaluated preprocessing time and model accuracy for the node classification task. The results are shown in Table 2. According to the new results, YOSO can achieve higher accuracy and much lower preprocessing time (12X faster than GCond and 6X faster than GC-SNTK.) comapred to the graph condensation-based scheme.

Table 2: Comparison of preprocessing time and model accuracy on the ogbn-arxiv dataset

| Dataset | ogbn-arxiv | | |
|---|---|---|---|
| Schemes | GCond | GC-SNTK | YOSO |
| Preprocessing Time (s) | 20615.6 | 11066.89 | 1643.32 |
| Model Accuracy (Metric: Accuracy) | 0.6172 | 0.6219 | 0.7169 |

### B.0.2 YOSO V.S. LINEARIZATION

To compare YOSO with linearization schemes, we conducted the following experiments: SIGN (Frasca et al., 2020) and iSVD (Abu-El-Haija et al., 2021) were selected as baselines. We compared the total training time and model accuracy on the ogbn-products and ogbn-arxiv datasets. We use iSVD-best to represent the version of this baseline with the highest accuracy, i.e., $iSVD_{250} + dropout(LR) + dropout(\hat{M}_{LR}^{(NC)}) + finetune\mathbf{H}$ in the paper. The results are shown in Table 3. From the results in the table, it can be observed that for SIGN, its accuracy on the

Table 3: Comparison between YOSO with linearization schemes.

| Dataset | ogbn-products | | | | | ogbn-arxiv | | |
|---|---|---|---|---|---|---|---|---|
| Schemes | SIGN-2 | SIGN-4 | SIGN-6 | SIGN-8 | YOSO | iSVD | iSVD-best | YOSO |
| Total Training Time (s) | 421.79 | 584.07 | 831.94 | 1052.96 | 499.02 | 9.94 | 982.12 | 10.74 |
| Model Accuracy (Metric: Accuracy) | 0.761 | 0.778 | 0.776 | 0.783 | 0.788 | 0.685 | 0.746 | 0.72 |

ogbn-products dataset does not exceed that of YOSO (0.788). While SIGN-2 achieves a total training time that is 18.3% lower than YOSO, its accuracy drops by 2.7%. A similar trend is observed for the iSVD baseline. The low-accuracy version of iSVD reduces total training time by 8% but suffers an accuracy drop of 4%. In contrast, the high-accuracy version, iSVD-best, increases total training time by 91X compared to YOSO, with only a 0.014 improvement in accuracy.

### B.0.3 LAYER-WISE $\Phi$ V.S. UNIVERSAL $\Phi$

A notable drawback of using layer-wise $\Phi$ is that its computational cost is disproportionately high compared to the improvement it brings in model accuracy. Specifically, to make this statement more clearly, we conducted the following experiments: on the datasets ogbn-arxiv and ogbl-ppa, we compared the total training time and model accuracy when using layer-wise $\Phi$ and universal $\Phi$. The results are shown in the Table 4. As seen in the table, across two different learning tasks, the

Table 4: Total training time and model accuracy on different types of $\Phi$

| Dataset | ogbn-arxiv | | ogbl-ppa | |
|---|---|---|---|---|
| Type of $\Phi$ | Layer-wise | Universal | Layer-wise | Universal |
| Total Training Time (s) | 59.22 | 10.93 | 145.5 | 21.46 |
| Model Accuracy | 0.73 | 0.727 | 0.2254 | 0.2235 |

layer-wise based scheme increases the total training by 5X (ogbn-arxiv) and 7X (ogbl-ppa), yielding only a marginal accuracy improvement at the 0.001 level compared to the universal-based scheme. Therefore, using a universal $\Phi$ does not lead to a significant reduction in accuracy.

## C DETAILS ABOUT EXPERIMENTS

### C.1 HARDWARE AND SOFTWARE CONFIGURATION

We evaluate all baselines and our design on a Linux Desktop running Ubuntu 18.04.6 LTS, equipped with an NVIDIA GTX 1060Ti (6GB memory) using CUDA version 11.8 and PyTorch version 2.0.0. The system features a AMD Ryzen 5 5500 CPU with 64 GB DDR4 RAM, and the Python version used is 3.9.0.

### C.2 DATASETS

**Data splitting**: We adopt strategies consistent with previous works (Hamilton et al., 2017; Hu et al., 2020). Specifically, for the Reddit dataset, we follow the data splitting used in GraphSage (Hamilton et al., 2017), and for the OGB series (ogbn and ogbl), we maintain the splitting described in (Hu et al., 2020).

The basic summary information of the datasets we use is provided in Table 5, and detailed descriptions are as follows:

Table 5: Statistics and metrics of the dataset

| Dataset | | #Node | #Edge | #Dim. | Metric |
|---|---|---|---|---|---|
| **Node Property Prediction** | ogbn-arxiv | 169,343 | 1,166,243 | 128 | Accuracy |
| | Reddit | 232,965 | 11,606,919 | 602 | Mirco-F1 |
| | ogbn-products | 2,449,029 | 61,859,140 | 100 | Accuracy |
| **Link Property Prediction** | ogbl-ppa | 576,289 | 30,326,273 | 128 | Hits@100 |
| | ogbl-citation2 | 2,927,963 | 30,561,187 | 128 | MRR |

Table 6: Baselines and their public available source code link

| Method | Available Link |
|---|---|
| GraphSage | https://github.com/williamleif/graphsage-simple |
| VR-GCN | https://github.com/THUDM/cogdl/tree/master/examples/VRGCN |
| FastGCN | https://github.com/gmancino/fastgcn-pytorch |
| AS-GCN | https://github.com/Gkunnan97/FastGCN_pytorch |
| LADIES | https://github.com/acbull/LADIES |
| Cluster-GCN | https://github.com/benedekrozemberczki/ClusterGCN |
| GraphSAINT | https://github.com/GraphSAINT/GraphSAINT |

**ogbn-arxiv**: This dataset is a directed citation network of Computer Science (CS) arXiv papers from the Microsoft Academic Graph (MAG) (Wang et al., 2020). Each node represents a paper, with directed edges indicating citations. The task is to classify unlabeled papers into primary categories using labeled papers and node features, which are derived by averaging word2vec embeddings (Mikolov et al., 2013) of paper titles and abstracts.

**Reddit**: Originally from GraphSage (Hamilton et al., 2017), this Reddit dataset is a post-to-post graph where each node represents a post, and edges indicate shared user comments. The task is to classify posts into communities using GloVe word vectors (Pennington et al., 2014) from post titles and comments, along with features such as post scores and comment counts.

**ogbn-products**: This undirected, unweighted graph represents an Amazon product co-purchasing network, where nodes are products and edges indicate frequent co-purchases. Node features are derived from bag-of-words features of product descriptions, reduced to 100 dimensions via Principal Component Analysis (Dunteman, 1989).

**ogbl-ppa**: This undirected, unweighted graph has nodes representing proteins from 58 species, with edges indicating biologically meaningful associations. Each node features a 58-dimensional one-hot vector for the protein's species. The task is to predict new association edges, evaluated by ranking positive test edges over negative ones.

**ogbl-citation2**: This dataset is a directed graph representing a citation network among a subset of papers from Microsoft Academic Graph (MAG), similar to ogbn-arxiv. For each source paper, two references are randomly removed, and the task is to rank these missing references above 1,000 randomly selected negative references, which are sampled from all papers not cited by the source paper.

## C.3 BASELINES AND IMPLEMENTATION

Table 6 presents the baselines used in this paper along with their publicly available source code links. Since some baselines were not originally implemented in PyTorch, we standardized the framework for fair comparison. If a PyTorch version involved the original authors, we selected that source code (e.g., FastGCN (Chen et al., 2018)). Otherwise, we chose the most popular implementation based on the number of stars. Notably, the repository linked for AS-GCN (Huang et al., 2018) in the table includes implementations of both FastGCN and AS-GCN, but we only used the AS-GCN version, while the FastGCN implementation was taken from the source listed in the table.

**YOSO's Implementation**: The base code of YOSO[1] is built on GCN (Kipf & Welling, 2016), with the link available at https://github.com/tkipf/pygcn. The sampling stage in YOSO occurs on the CPU and main memory since it involves calculations related to the entire feature matrix and the regularized Laplacian matrix. After sampling, the relevant data is migrated to GPU memory

---

[1]https://anonymous.4open.science/r/YOSO-B49B

Table 7: Node classification hyperparamter setting for baselines and YOSO on different datasets.

|  | ogbn-arxiv | Reddit | ogbn-products |
|---|---|---|---|
| GraphSage | 25&10 / Adam / 0.7 | 25&10 / Adam / 0.01 | 50&20 / Adam / 0.01 |
| VR-GCN | 8 / Adam / 0.01 | 16 / Adam / 0.01 | 32 / Adam / 0.01 |
| FastGCN | 64 / Adam / 0.01 | 128 / Adam / 0.001 | 256 / Adam / 0.001 |
| AS-GCN | 128 / Adam / 0.001 | 512 / Adam / 0.01 | 1000 / Adam / 0.01 |
| LADIES | 64 / Adam / 0.001 | 128 / Adam / 0.001 | 256 / Adam / 0.001 |
| Cluster-GCN | - / Adam / 0.01 | - / Adam / 0.005 | - / Adam / 0.005 |
| GraphSAINT-EG | 300 / Adam / 0.01 | 600 / Adam / 0.01 | 4000 / Adam / 0.01 |
| GraphSAINT-RW | 4000 / Adam / 0.01 | 8000 / Adam / 0.01 | 10000 / Adam / 0.01 |
| YOSO | 128 / Adam / 0.01 | 256 / Adam / 0.01 | 512 / Adam / 0.01 |

Table 8: Link prediction hyperparamter setting for baselines and YOSO on different datasets.

|  | ogbl-ppa | ogbl-citation2 |
|---|---|---|
| GraphSage | 25&10 / Adam / 0.7 | 50&20 / Adam / 0.01 |
| VR-GCN | 8 / Adam / 0.01 | 32 / Adam / 0.01 |
| FastGCN | 64 / Adam / 0.01 | 256 / Adam / 0.001 |
| AS-GCN | 128 / Adam / 0.001 | 1000 / Adam / 0.01 |
| LADIES | 64 / Adam / 0.001 | 256 / Adam / 0.001 |
| Cluster-GCN | - / Adam / 0.01 | - / Adam / 0.005 |
| GraphSAINT-EG | 300 / Adam / 0.01 | 4000 / Adam / 0.01 |
| GraphSAINT-RW | 4000 / Adam / 0.01 | 10000 / Adam / 0.01 |
| YOSO | 128 / Adam / 0.01 | 512 / Adam / 0.01 |

for computation. Throughout the training process, multiple data exchanges occur between main memory and GPU memory, such as in link prediction tasks where node embeddings need to be updated.

**Modification:** All baselines support updating node embeddings and performing node classification tasks. For node classification, if a baseline did not originally use the cross-entropy loss function, we adjusted it to adopt this loss function. For the link prediction task, the following loss function is applied:

$$
\mathcal{L} = \frac{1}{N^+} \sum_{(i,j) \in E^+} \left( 1 - \frac{\mathbf{h}_i^{(L)} \cdot \mathbf{h}_j^{(L)}}{\|\mathbf{h}_i^{(L)}\| \|\mathbf{h}_j^{(L)}\|} \right) + \frac{1}{N^-} \sum_{(i,j) \in E^-} \max \left( 0, \gamma - \left( 1 - \frac{\mathbf{h}_i^{(L)} \cdot \mathbf{h}_j^{(L)}}{\|\mathbf{h}_i^{(L)}\| \|\mathbf{h}_j^{(L)}\|} \right) \right)
$$

where $N^+$ and $N^-$ represent the number of positive and negative samples, respectively, and $E^+$ and $E^-$ denote the sets of positive and negative edges. The parameter $\gamma$ is a hyperparameter, set to 0.5 in this study. As the ogbl-ppa and ogbl-citation2 datasets provide corresponding negative edges by default, we used these pre-defined negative edges for our calculations.

## C.4 Hyper-parameter Setting

The hyperparameter settings for both YOSO and the baselines are provided in Table 7 and Table 8 for node classification and link prediction datasets, respectively. All experiments were conducted using a two-layer GCN with official configurations. When certain parameters were not clearly specified in some papers, we fine-tuned them for optimal accuracy. The recorded hyperparameters include the sampling size (per node/layer/subgraph), the optimizer, and the learning rate. For YOSO, the sampling size is denoted as $M$; for example, on the ogbl-ppa dataset (Table 8), $M = 128$.

# D COMPUTATION AND PROOF

## D.1 GRADIENT COMPUTATION

### D.1.1 COMPUTATION OF $\nabla_{\Theta}\mathcal{L}$:

$$\nabla_{\Theta}\mathcal{L} = \alpha\nabla_{\Theta}\mathcal{L}_{recon} + \beta\nabla_{\Theta}\mathcal{L}^{\Theta}_{GNN}(\mathbf{Z}) = \frac{\partial\mathcal{L}_{recon}}{\partial\mathbf{Z}} \cdot \frac{\partial\mathbf{Z}}{\partial\Theta} + \frac{\partial\mathcal{L}^{\Theta}_{GNN}(\mathbf{Z})}{\partial\mathbf{Z}} \cdot \frac{\partial\mathbf{Z}}{\partial\Theta}$$

- $\frac{\partial\mathcal{L}_{recon}}{\partial\mathbf{Z}} = (\mathbf{Z} - \mathbf{\Phi}\mathbf{U}\hat{\mathbf{H}}^{(L)})$

- Consider the $g^{(L)}$ which is the gradient at the output layer, and we have $g^{(L)} = \frac{\partial\mathcal{L}_{recon}}{\partial\mathbf{Z}} \odot \sigma^{'}(\mathbf{S}^{(L)})$ where $\odot$ denotes element-wise multiplication, $\sigma^{'}(\mathbf{S}^{(L)})$ is the derivation of the activation function at layer $L$ and $\mathbf{S}^{(L)}$ is the pre-activation input at layer $L$. Therefore, for $l = L, L-1, ..., 1$, we have $g^{(l-1)} = \nabla_{\mathbf{W}^{(L)}}\mathcal{L}_{recon} \odot \sigma^{'}(\mathbf{S}^{(l-1)}) = (\mathbf{\Phi}\hat{\mathbf{A}}\mathbf{W}^{(l)})^T g^{(l)} \odot \sigma^{'}(\mathbf{S}^{(l-1)})$. By iteratively executing this process, we can obtain $\frac{\partial\mathbf{Z}}{\partial\Theta}$

- $\frac{\partial\mathcal{L}^{\Theta}_{GNN}}{\partial\mathbf{Z}}$ depends on the specific loss function used.

### D.1.2 COMPUTATION OF $\nabla_{\mathbf{U}}\mathcal{L}$

$$\nabla_{\mathbf{U}}\mathcal{L} = \alpha\nabla_{\mathbf{U}}\mathcal{L}_{recon} + \beta\nabla_{\mathbf{U}}\mathcal{L}^{\Theta}_{GNN}(\mathbf{Z}) = \alpha\nabla_{\mathbf{U}}\mathcal{L}_{recon} + \beta(\frac{\partial\mathcal{L}^{\Theta}_{GNN}(\mathbf{Z})}{\partial\mathbf{Z}} \cdot \frac{\partial\mathbf{Z}}{\partial\mathbf{U}})$$

- $\nabla_{\mathbf{U}}\mathcal{L}_{recon} = -\mathbf{\Phi}^T(\mathbf{Z} - \mathbf{\Phi}\mathbf{U}\hat{\mathbf{H}}^{(L)})(\hat{\mathbf{H}}^{(L)})^T$

- As in Section D.1.1, $\frac{\partial\mathcal{L}^{\Theta}_{GNN}}{\partial\mathbf{Z}}$ depends on specific loss function and easy to compute.

- For $\frac{\partial\mathbf{Z}}{\partial\mathbf{U}}$, it need to be computed recursively. Since $\mathbf{T}^{(0)} = \mathbf{\Phi}\mathbf{U}\hat{\mathbf{X}}$, $\frac{\partial\mathbf{T}^{(0)}}{\partial\mathbf{U}} = \mathbf{\Phi}\hat{\mathbf{X}}$. The gradient propagates from $\mathbf{Z}$ back to $\mathbf{U}$: $\nabla_{\mathbf{U}}\mathcal{L}^{\Theta}_{GNN}(\mathbf{Z}) = (\frac{\partial\mathcal{L}^{\Theta}_{GNN}(\mathbf{Z})}{\partial\mathbf{Z}} \cdot \frac{\partial\mathbf{Z}}{\partial\mathbf{T}^{(L-1)}} \cdots \frac{\partial\mathbf{T}^{(1)}}{\partial\mathbf{T}^{(0)}} \cdot \frac{\partial\mathbf{T}^{(0)}}{\partial\mathbf{U}})$. As we know that $\mathbf{T}^{(l)} = \sigma(\mathbf{S}^{(l)})$ and $\mathbf{S}^{(l)} = \mathbf{\Phi}\hat{\mathbf{A}}\mathbf{W}^{(l)}\mathbf{T}^{(l-1)}$, therefore $\frac{\partial\mathbf{T}^{(l-1)}}{\partial\mathbf{T}^{(l-1)}} = (\mathbf{\Phi}\hat{\mathbf{A}}\mathbf{W}^{(l)})^T\text{diag}(\sigma^{'}(\mathbf{S}^{(l)}))$

### D.1.3 COMPUTATION OF $\nabla_{\hat{\mathbf{H}}^{(L)}}\mathcal{L}$

$\nabla_{\hat{\mathbf{H}}^{(L)}}\mathcal{L} = \alpha\nabla_{\hat{\mathbf{H}}^{(L)}}\mathcal{L}_{recon} = -\mathbf{U}^T\mathbf{\Phi}^T(\mathbf{Z} - \mathbf{\Phi}\mathbf{U}\hat{\mathbf{H}}^{(L)}) + \lambda\partial\|\hat{\mathbf{H}}^{(L)}\|_{2,1}$ where $\partial\|\hat{\mathbf{H}}^{(L)}\|_{2,1}$ is the subgradient of the $l_{2,1}$ norm and computed as $(\partial\|\hat{\mathbf{H}}^{(L)}\|_{2,1})_i = \frac{\hat{\mathbf{H}}^{(L)}_{i,:}}{\|\hat{\mathbf{H}}^{(L)}_{i,:}\|_2}$ if and only if $\hat{\mathbf{H}}^{(L)}_{i,:} \neq 0$, otherwise, $(\partial\|\hat{\mathbf{H}}^{(L)}\|_{2,1})_i = 0$

## D.2 FULL RANK OF $\mathbf{\Phi}$

**Theorem 1:** Let $\hat{\mathbf{S}} \in \mathbb{R}^{M \times N}$ be a binary sampling matrix derived from the graph's structure, where each entry $\hat{\mathbf{S}}_{i,j} \in \{0, 1\}$ and each row has at least one non-zero entry. Let $\mathbf{\Sigma} \in \mathbb{R}^{M \times N}$ be a random matrix with entries drawn independently from a continuous probability distribution. Define $\mathbf{\Phi} = \hat{\mathbf{S}} \otimes \mathbf{\Sigma}$, where $\otimes$ denotes element-wise multiplication. Then, with probability 1, the matrix $\mathbf{\Phi}$ has full row rank $M$.

**Proof:** First, we know that the structure of $\mathbf{\Phi}$ satisfies the following conditions:

- Each entry of $\mathbf{\Phi}$ is given by $\mathbf{\Phi}_{i,j} = \hat{\mathbf{S}}_{i,j} \cdot \mathbf{\Sigma}_{i,j}$.

- The $i$-th row of $\mathbf{\Phi}$ is $\mathbf{\Phi}_{i,:} = \hat{\mathbf{S}}_{i,:} \otimes \mathbf{\Sigma}_{i,:}$.

- Non-zero entries in $\mathbf{\Phi}_{i,:}$ correspond to positions where $\hat{\mathbf{S}}_{i,j} = 1$.

Assume there exist scalars $c_1, c_2, \ldots, c_M$, not all zero, such that $\sum_{i=1}^M c_i\mathbf{\Phi}_{i,:} = \mathbf{0}$. This implies that for each $j = 1, \ldots, N$, we have $\sum_{i=1}^M c_i\hat{\mathbf{S}}_{i,j}\mathbf{\Sigma}_{i,j} = 0$. Let $I_j = \{i \mid \hat{\mathbf{S}}_{i,j} = 1\}$; then $\sum_{i \in I_j} c_i\mathbf{\Sigma}_{i,j} = 0$.

Since the $\mathbf{\Sigma}_{i,j}$ values are independently drawn from continuous distributions, the probability that this equation holds for any non-zero set of $\{c_i\}$ is zero unless all $c_i$ in $I_j$ are zero. Therefore, for the equation to be valid, $c_i = 0$ for all $i$ where $\hat{\mathbf{S}}_{i,j} = 1$.

As each row $i$ contains at least one entry with $\hat{\mathbf{S}}_{i,j} = 1$, it follows that $c_i = 0$ for all $i$. This contradicts the assumption that not all $c_i$ are zero. Hence, the only solution is $c_i = 0$ for all $i$, indicating that the rows of $\mathbf{\Phi}$ are linearly independent with probability 1. Thus, $\text{rank}(\mathbf{\Phi}) = M$ with probability 1.

### D.3 Sampling Matrix $\mathbf{\Phi}$ and RIP

**Theorem 2:** Let $\hat{\mathbf{S}} \in \mathbb{R}^{M \times N}$ be a selection matrix derived from the graph's structure, where each entry $\hat{\mathbf{S}}_{i,j} \in \{0,1\}$ indicates whether node $j$ is included in the $i$-th measurement. Let $\mathbf{\Sigma} \in \mathbb{R}^{M \times N}$ be a matrix whose entries $\mathbf{\Sigma}_{i,j}$ are independent sub-Gaussian random variables with mean zero and variance $\frac{1}{g(j)}$, where $g(j) > 0$. Define the sampling matrix $\mathbf{\Phi} = \hat{\mathbf{S}} \otimes \mathbf{\Sigma}$, where $\otimes$ denotes element-wise multiplication. Then, for any $0 < \delta_k < 1$, there exists a constant $c > 0$ such that if $M \geq c \cdot k \log\left(\frac{N}{k}\right)$, then with probability at least $1 - e^{-cM}$, the matrix $\mathbf{\Phi}\mathbf{U}$ satisfies the Restricted Isometry Property (RIP) of order $k$ with constant $\delta_k$; that is, for all $\hat{\mathbf{H}} \in \mathbb{R}^{N \times d}$ with $\|\hat{\mathbf{H}}\|_{0,\text{row}} \leq k$,

$$(1 - \delta_k)\|\hat{\mathbf{H}}\|_F^2 \leq \|\mathbf{\Phi}\mathbf{U}\hat{\mathbf{H}}\|_F^2 \leq (1 + \delta_k)\|\hat{\mathbf{H}}\|_F^2.$$

**Proof:** To demonstrate that $\mathbf{\Phi}\mathbf{U}$ satisfies the Restricted Isometry Property (RIP) of order $k$ with high probability, we consider $\mathbf{\Phi}\mathbf{U}\hat{\mathbf{H}} = (\hat{\mathbf{S}} \otimes \mathbf{\Sigma})\mathbf{U}\hat{\mathbf{H}}$. For each row $i$ and column $r$, the entry $(\mathbf{\Phi}\mathbf{U}\hat{\mathbf{H}})_{i,r}$ can be expressed as $\sum_{j=1}^{N} \hat{\mathbf{S}}_{i,j}\mathbf{\Sigma}_{i,j}(\mathbf{U}\hat{\mathbf{H}})_{j,r}$. This sum only involves terms where $\hat{\mathbf{S}}_{i,j} = 1$. Therefore, $(\mathbf{\Phi}\mathbf{U}\hat{\mathbf{H}})_{i,r} = \sum_{j \in \mathcal{S}_i} \mathbf{\Sigma}_{i,j}(\mathbf{U}\hat{\mathbf{H}})_{j,r}$, where $\mathcal{S}_i = \{j \mid \hat{\mathbf{S}}_{i,j} = 1\}$.

The variables $\mathbf{\Sigma}_{i,j}$ are independent sub-Gaussian random variables with mean zero and variance $\frac{1}{g(j)}$. Therefore, the expectation of $\|\mathbf{\Phi}\mathbf{U}\hat{\mathbf{H}}\|_F^2$ can be computed as follows:

$$\mathbb{E}\left[\|\mathbf{\Phi}\mathbf{U}\hat{\mathbf{H}}\|_F^2\right] = \sum_{i=1}^{M} \sum_{r=1}^{d} \mathbb{E}\left[\left(\sum_{j \in \mathcal{S}_i} \mathbf{\Sigma}_{i,j}(\mathbf{U}\hat{\mathbf{H}})_{j,r}\right)^2\right]$$

Expanding this and leveraging the independence of $\mathbf{\Sigma}_{i,j}$, we have:

$$\mathbb{E}\left[\left(\sum_{j \in \mathcal{S}_i} \mathbf{\Sigma}_{i,j}(\mathbf{U}\hat{\mathbf{H}})_{j,r}\right)^2\right] = \sum_{j \in \mathcal{S}_i} \mathbb{E}\left[\mathbf{\Sigma}_{i,j}^2\right]\left((\mathbf{U}\hat{\mathbf{H}})_{j,r}\right)^2$$

Since $\mathbb{E}[\mathbf{\Sigma}_{i,j}^2] = \frac{1}{g(j)}$, the expectation simplifies to:

$$\mathbb{E}\left[\|\mathbf{\Phi}\mathbf{U}\hat{\mathbf{H}}\|_F^2\right] = \sum_{i=1}^{M} \sum_{j \in \mathcal{S}_i} \frac{1}{g(j)} \sum_{r=1}^{d}\left((\mathbf{U}\hat{\mathbf{H}})_{j,r}\right)^2$$

If we assume $p(j) = \frac{g(j)}{G}$, where $G = \sum_{j=1}^{N} g(j)$ serves as a normalization factor, the expected measurement count for each node $j$ is $Mp(j) = M\frac{g(j)}{G}$. Thus:

$$\mathbb{E}\left[\|\mathbf{\Phi}\mathbf{U}\hat{\mathbf{H}}\|_F^2\right] = \sum_{j=1}^{N} M\frac{g(j)}{G} \cdot \frac{1}{g(j)}\|(\mathbf{U}\hat{\mathbf{H}})_{j,:}\|_2^2 = \frac{M}{G}\|\mathbf{U}\hat{\mathbf{H}}\|_F^2$$

By setting $G = M$, we have:

$$\mathbb{E}\left[\|\mathbf{\Phi}\mathbf{U}\hat{\mathbf{H}}\|_F^2\right] = \|\hat{\mathbf{H}}\|_F^2$$

Now, define $Z_{i,r} = \sum_{j \in \mathcal{S}_i} \mathbf{\Sigma}_{i,j}(\mathbf{U}\hat{\mathbf{H}})_{j,r}$, which are sub-Gaussian random variables. Applying Bernstein's inequality, we obtain:

$$\mathbb{P}\left(\left|\|\mathbf{\Phi}\mathbf{U}\hat{\mathbf{H}}\|_F^2 - \|\hat{\mathbf{H}}\|_F^2\right| \geq \delta_k\|\hat{\mathbf{H}}\|_F^2\right) \leq 2\exp\left(-c \cdot \frac{\delta_k^2\|\hat{\mathbf{H}}\|_F^4}{\sum_{i,r} \sigma_{i,r}^2}\right)$$

where $\sigma_{i,r}^2 = \sum_{j \in \mathcal{S}_i} \frac{1}{g(j)} \left( (\mathbf{U}\hat{\mathbf{H}})_{j,r} \right)^2$. By bounding the total variance, we conclude that the probability of RIP failing is very low. This confirms that $\mathbf{\Phi}\mathbf{U}$ satisfies the RIP for all sparse $\hat{\mathbf{H}}$ with $\|\hat{\mathbf{H}}\|_{0,\text{row}} \le k$ with high probability.

## D.4 Error Bound

**Theorem 3:** Let $\mathbf{H}^{(L)}$ be the output embeddings obtained by the standard GNN computation with full reconstruction at each layer as per Equation (6). Let $\tilde{\mathbf{H}}^{(L)}$ be the output embeddings obtained by Algorithm 1, which performs sampling once at the input layer and reconstructs only at the output layer. Assume that the activation function $\sigma$ is Lipschitz continuous with Lipschitz constant $L_\sigma$, and the sampling matrix $\mathbf{\Phi}\mathbf{U}$ satisfies the Restricted Isometry Property (RIP) of order $k$ with constant $\delta_k$ (i.e., $0 < \delta_k < 1$). Then, the error between $\tilde{\mathbf{H}}^{(L)}$ and $\mathbf{H}^{(L)}$ can be bounded as:

$$\left\| \tilde{\mathbf{H}}^{(L)} - \mathbf{H}^{(L)} \right\|_F \le \left( \frac{L_\sigma}{1 - \delta_k} \right)^L \|\mathbf{E}\|_F \,,$$

where $\mathbf{E} = \mathbf{Z} - \mathbf{\Phi}\mathbf{U}\hat{\mathbf{H}}^{(L)}$ is the reconstruction error at the output layer, and $L$ is the number of layers in the GNN.

**Proof:** We aim to bound the error $\left\| \tilde{\mathbf{H}}^{(L)} - \mathbf{H}^{(L)} \right\|_F$ between the output embeddings of the standard GNN computation and those obtained by Algorithm 1.

Assume the activation function $\sigma$ is Lipschitz continuous with a constant $L_\sigma$, such that

$$\|\sigma(\mathbf{X}) - \sigma(\mathbf{Y})\|_F \le L_\sigma \|\mathbf{X} - \mathbf{Y}\|_F \quad \forall \mathbf{X}, \mathbf{Y}.$$

Further, let the sampling matrix $\mathbf{\Phi}\mathbf{U}$ satisfy the RIP of order $k$ with constant $\delta_k$, meaning

$$(1 - \delta_k) \left\| \hat{\mathbf{H}} \right\|_F^2 \le \left\| \mathbf{\Phi}\mathbf{U}\hat{\mathbf{H}} \right\|_F^2 \le (1 + \delta_k) \left\| \hat{\mathbf{H}} \right\|_F^2,$$

for all $\hat{\mathbf{H}}$ with $\left\| \hat{\mathbf{H}} \right\|_{0,\text{row}} \le k$. We also have $\mathbf{H}^{(l)} = \mathbf{U}\hat{\mathbf{H}}^{(l)}$, where $\hat{\mathbf{H}}^{(l)}$ has at most $k$ non-zero rows.

We will prove by induction on $l = 1, 2, \ldots, L$ that

$$\left\| \tilde{\mathbf{H}}^{(l)} - \mathbf{H}^{(l)} \right\|_F \le \left( \frac{L_\sigma}{1 - \delta_k} \right)^l \left\| \tilde{\mathbf{H}}^{(0)} - \mathbf{H}^{(0)} \right\|_F.$$

For the base case $l = 0$, at the input layer, we have $\tilde{\mathbf{H}}^{(0)} = \mathbf{U}\hat{\mathbf{X}}$ and $\mathbf{H}^{(0)} = \mathbf{X}$. The initial error $\left\| \tilde{\mathbf{H}}^{(0)} - \mathbf{H}^{(0)} \right\|_F$ is assumed.

Assume that for some $l \ge 0$,

$$\left\| \tilde{\mathbf{H}}^{(l)} - \mathbf{H}^{(l)} \right\|_F \le \left( \frac{L_\sigma}{1 - \delta_k} \right)^l \left\| \tilde{\mathbf{H}}^{(0)} - \mathbf{H}^{(0)} \right\|_F.$$

We aim to show that

$$\left\| \tilde{\mathbf{H}}^{(l+1)} - \mathbf{H}^{(l+1)} \right\|_F \le \left( \frac{L_\sigma}{1 - \delta_k} \right)^{l+1} \left\| \tilde{\mathbf{H}}^{(0)} - \mathbf{H}^{(0)} \right\|_F.$$

For Algorithm 1, $\tilde{\mathbf{T}}^{(l)} = \sigma \left( \mathbf{\Phi}\hat{\mathbf{A}}\mathbf{W}^{(l+1)}\tilde{\mathbf{T}}^{(l-1)} \right)$. At the output layer $l = L$, we perform reconstruction:

$$\tilde{\mathbf{H}}^{(L)} = \mathbf{U}\hat{\mathbf{H}}^{(L)},$$

where $\hat{\mathbf{H}}^{(L)}$ is obtained by solving

$$\min_{\hat{\mathbf{H}}^{(L)}} \frac{1}{2} \left\| \mathbf{Z} - \mathbf{\Phi}\mathbf{U}\hat{\mathbf{H}}^{(L)} \right\|_F^2 + \lambda \left\| \hat{\mathbf{H}}^{(L)} \right\|_{2,1},$$

with $\mathbf{Z} = \tilde{\mathbf{T}}^{(L)}$. Due to the optimization and the RIP condition, we have

$$\left\| \hat{\mathbf{H}}^{(L)} - \hat{\mathbf{H}}^{(L)}_{\text{true}} \right\|_F \leq C_{\text{rec}} \left\| \mathbf{E} \right\|_F ,$$

where $\hat{\mathbf{H}}^{(L)}_{\text{true}}$ is the true sparse representation of $\mathbf{H}^{(L)}$, and $C_{\text{rec}} = \frac{2\delta_k}{1-\delta_k}$. Since $\mathbf{U}$ is orthonormal,

$$\left\| \tilde{\mathbf{H}}^{(L)} - \mathbf{H}^{(L)} \right\|_F = \left\| \hat{\mathbf{H}}^{(L)} - \hat{\mathbf{H}}^{(L)}_{\text{true}} \right\|_F ,$$

implying

$$\left\| \tilde{\mathbf{H}}^{(L)} - \mathbf{H}^{(L)} \right\|_F \leq \frac{2\delta_k}{1-\delta_k} \left\| \mathbf{E} \right\|_F .$$

Given the Lipschitz continuity of $\sigma$, the error accumulates multiplicatively through $L$ layers:

$$\left\| \tilde{\mathbf{H}}^{(L)} - \mathbf{H}^{(L)} \right\|_F \leq \left( \frac{L_\sigma}{1-\delta_k} \right)^L \left\| \tilde{\mathbf{H}}^{(0)} - \mathbf{H}^{(0)} \right\|_F .$$

If the initial error $\left\| \tilde{\mathbf{H}}^{(0)} - \mathbf{H}^{(0)} \right\|_F = 0$, the primary source of error is from the reconstruction at the output layer, yielding

$$\left\| \tilde{\mathbf{H}}^{(L)} - \mathbf{H}^{(L)} \right\|_F \leq \left( \frac{L_\sigma}{1-\delta_k} \right)^L \left\| \mathbf{E} \right\|_F .$$

