# OpenReview forum: "YOSO: You-Only-Sample-Once via Compressed Sensing for Graph Neural Network Training"
_ICLR.cc/2025/Conference — Submitted to ICLR 2025_

### Official Review · Reviewer_VRqU · 2024-10-28

**Soundness:** 2
**Presentation:** 2
**Contribution:** 2
**Rating:** 6
**Confidence:** 3

**Summary:**

This paper introduces YOSO, a novel algorithm for efficient GNN training using a compressed sensing approach. GNNs are crucial for analyzing structured data, but existing sampling methods often introduce significant computational overhead. YOSO addresses this by sampling nodes only once at the input layer and using a nearly lossless reconstruction at the output layer for each epoch. This method reduces training time by about 75% compared to state-of-the-art techniques while maintaining high accuracy.

**Strengths:**

1. The concept of "sampling only once" presented in this paper is really interesting and novel, offering a fresh perspective on GNN sampling.
2. The author provides theoretical support to guarantee the effectiveness of YOSO, which is very convincing.
3. The author provides code to ensure the reproducibility of the paper's results.

**Weaknesses:**

1. According to Formula 1, the generation of $\mathbf{T}$ is closely related to the feature matrix $\mathbf{H}$. However, in YOSO, $\mathbf{U}$ is randomly initialized, and the generation of $\mathbf{\Phi}$ is only related to the graph structure. Does this approach overlook too much feature information?
2. The methodology section of the paper lacks a description of critical steps. The resulting representations $\mathbf{Z}$  do not have the scale as the input matrix $\mathbf{X}$. So, how should the nodes in $\mathbf{X}$ correspond to the representations in $\mathbf{Z}$? How can we obtain the representations for all nodes in the entire graph? What nodes' loss does $L_{GNN}(\mathbf{Z})$ calculate? Does this mean that it is necessary to sample nodes from the training set?
3. Some important related works are missing. The paper only discusses related work on graph sampling, while relevant works on Compressed Sensing are not included. What are the objectives and common practices of this technique? Why can it be applied to the GNN sampling problem? These questions need further discussion.
4. The mathematical notation in the paper is confusing, making it difficult to understand. For example, in line 155, $\hat{\mathbf{H}}$ should be $\hat{\mathbf{H}}^{(l)}$; in line 159, $\mathbf{U}^{\top}$ should be ${\mathbf{U}^{(l)}}^\top$; and in Equation 2, $\min$ should be $\arg \min$.

**Questions:**

Please refer to the points I mentioned in the weakness part.

---

> ### Author Response · Authors · 2024-11-21
>
> Dear Reviewer VRqU,
> Thank you for your valuable comments. Please find our detailed response blow.
>
> ## Weakness
> ### 1. Overlook too much feature information
> Thank you for pointing this out. Actually, this approach does not overlook feature information. Equation (1) (i.e., $\mathbf{T} = \mathbf{\Phi}\mathbf{U}\hat{\mathbf{H}}$) and the following computations, including sampling, are all performed based on $\mathbf{T}$. Finally, we fully reconstruct the corresponding $\mathbf{H}$ from $\mathbf{T}$. In other words, YOSO fully utilizes the feature information during the training. This differs from some previous feature-dependent sampling methods, which calculate some measure of importance based on the feature information [1,2].
>
> ### 2. The shape of $\mathbf{Z}$ and input $\mathbf{X}$
> Assuming that node features/hidden representations/embeddings are arranged row-wise and the GNN has $L$ layers, the number of rows in $\mathbf{Z}$ will be much smaller than that in $\mathbf{X}$. In YOSO's design, $\mathbf{Z}$ is the result of compressing and sampling $\mathbf{H}^{(L)}$ (whose number of rows equals the number of nodes in the dataset). This compressing and sampling process starts from $\mathbf{X}$, as shown in Equation (1) of the paper, and all subsequent computations are performed within this compressed/sampled data domain.
> According to YOSO's design, $\mathbf{H}^{(L)}$ can be reconstructed from $\mathbf{Z}$ losslessly or nearly losslessly, thereby we obtain the embeddings for all nodes. Consequently, loss $L_{GNN}(\mathbf{Z})$ can be computed.
>
> ### 3. Related work and CS-GNN Sampling Relationship
> We have added a background description of compressed sensing (CS) in the related work (Section 2.2) of the revised paper and enhanced the explanation of how CS is applied to GNN sampling in Section 3 and Section 4. The relevant parts have been highlighted in blue.
>
> ### 4. Mathematical notation
> Thank you for your valuable feedback. We have fixed the inconsistency in mathematical notation in the revised version of the paper and highlighted the changes in blue.
>
>
>
>
> **We hope our responses have effectively addressed your concerns. If so, we kindly request that you consider updating your score. Should you have any additional questions or feedback, please don’t hesitate to reach out. We’d be more than happy to provide further clarification.**
>
>
>
>
>
> #### Reference
> [1] Chen, Jie, Tengfei Ma, and Cao Xiao. "Fastgcn: fast learning with graph convolutional networks via importance sampling." arXiv preprint arXiv:1801.10247 (2018).
> [2] Liu, Ziqi, et al. "Bandit samplers for training graph neural networks." Advances in Neural Information Processing Systems 33 (2020): 6878-6888.

---

> ### Comment · Reviewer_VRqU · 2024-11-23
>
> Thank you for the author's patient responses. I have thoroughly read all the author's replies as well as the feedback from other reviewers. The revisions in the new version of the paper are quite significant. However, specific explanations like your response to my Question 2 should also be included in the main text to enhance the paper's readability. In fact, almost all reviewers believe that the description of the main method needs improvement. I think Section 5 (especially 5.1) could be further refined.

---

> > ### Author Response · Authors · 2024-11-26
> > **Follow-up on Review Feedback**
> >
> > Dear Reviewer VRqU,
> >
> > Thank you for your valuable feedback on our submission. We truly appreciate the time and effort you have devoted to reviewing our work.
> >
> > We have carefully addressed the concerns raised in your review and updated the manuscript accordingly. As tomorrow is the final day for updates the manuscript, we would like to confirm whether the current version aligns with your expectations. If there are any remaining issues or further suggestions, we would be more than happy to make additional adjustments.
> >
> > If you find the updated version meets the standards, we would greatly appreciate it if you could consider revising the score to reflect the improvements made.
> >
> > Thank you once again for your guidance and support.

---

> > > ### Comment · Reviewer_VRqU · 2024-11-30
> > >
> > > I appreciate the efforts the authors have made in their responses. And I'm willing to increase the score by one point.

---

> > > > ### Author Response · Authors · 2024-12-01
> > > > **Thank You for Your Review, Ongoing Engagement and Updated Score**
> > > >
> > > > Dear Reviewer VRqU,
> > > >
> > > > Thank you for your thoughtful review and constructive suggestions, as well as for acknowledging the improvements in our paper. We truly appreciate your updated score and your ongoing engagement.

---

> ### Author Response · Authors · 2024-11-24
>
> Dear Reviewer VRqU,
>
> Thank you once again for your feedback regarding the paper's readability. Based on your suggestions, we have made additional revisions to the paper. Specifically, to make the description of the YOSO method (Section 5) clearer and more coherent, we have implemented the following changes:
> 1. **Integration of specific explanations/responses.** The weaknesses you mentioned, along with our responses, which were not included in the previous version of the paper, have now been addressed and incorporated into Section 5.2 of the current version (e.g., Weakness 2). Additionally, we have revised the mathematical notation to describe these issues more clearly.
> 2. Revised Section 5
>     (1). **Added Section 5.1.** Section 5.1 provides a high-level overview and explains the motivation behind YOSO's design, emphasizing the inputs and outputs of each component.
>     (2). **Revised Section 5.2 (formerly Section 5.1):** Current Section 5.2 elaborates on the details described in Current Section 5.1. Thus bridges the gap in understanding the YOSO method design by transitioning from a high-level description (Section 5.1) to a detailed explanation (Section 5.2).
>     (3). **Revisions to wording and phrasing.** We primarily refined Section 5.2 and Section 5.3 to make them more coherent and improve readability.
>
> All changes have been highlighted in blue and revised paper is uploaded.
>
> If you find the revised version meets the readability requirements, we sincerely hope you will consider updating your score. Should you have any additional questions or feedback, please don’t hesitate to reach out. We would be more than happy to provide further clarification.

---

### Official Review · Reviewer_9rsa · 2024-11-04

**Soundness:** 3
**Presentation:** 2
**Contribution:** 4
**Rating:** 6
**Confidence:** 3

**Summary:**

This paper presents YOSO (You-Only-Sample-Once), an efficient algorithm designed to streamline graph neural network (GNN) training while maintaining predictive accuracy in downstream tasks. YOSO addresses the high computational overhead in existing sampling methods by introducing a novel compressed sensing-based framework. In this framework, nodes are sampled once at the input layer and then losslessly reconstructed at the output layer each epoch, eliminating costly operations like orthonormal basis calculations and ensuring high-probability accuracy retention comparable to full-node sampling. Experimental evaluations demonstrate YOSO’s effectiveness in reducing GNN training time by approximately 75% on tasks such as node classification and link prediction, achieving accuracy levels similar to leading baselines. This approach positions YOSO as a resource-efficient alternative with potential for significant performance benefits in large-scale GNN training.

**Strengths:**

1. The paper introduces a fairly novel method to train the GNNs, which is a significant as per the details provided.
2. The mathematical representation provided is fair enough to make the readers understand.
3. The results and the studies are significant to show the efectiveness of the framework. As per the problem, large-scale datasets are considered for the study, which is also appreciable.

**Weaknesses:**

1. The work is evaluated for only two tasks, i.e., Node classification and link prediction, in which the link prediction results are also just comparable to the baselines.
2. The performance metrics considered is accuracy and loss only. There should me more like F1-score as there may be data bias in case of such datasets.

**Questions:**

1. According to the article, YOSO operates on sparse data rather than the original dataset. Could this approach lead to potential information loss?
2. If there is data bias, how will it handle the same?
3. As the framework is tested for just two tasks, will it be generic for all type of graph related downstream tasks?

---

> ### Author Response · Authors · 2024-11-21
>
> Dear Reviewer 9rsa,
> Thank you for your valuable comments. Please find our detailed response blow.
>
> ## Weakness
> ### 1. The number of evaluated tasks
> The reasons why we only evaluated and reported two graph learning tasks (node classification and link prediction) in the paper are as follows:
>
> As the introduction outlines, our work aims to "reveal the large overhead introduced by sampling in GNN training" closely related to tasks on a single large graph. As the size of a single graph increases, the search space during the sampling process also grows, leading to increased overhead for existing sampling algorithms. YOSO addresses this specific issue. In benchmark datasets such as OGB, tasks like node classification and link prediction often involve extremely large graphs, such as those in [OGBN](https://ogb.stanford.edu/docs/nodeprop/) and [OGBL](https://ogb.stanford.edu/docs/linkprop/) (tens or hundred GBs per graph). These tasks are particularly relevant for demonstrating the challenges our work addresses and the performance improvement lead by our work.
>
> In contrast, tasks such as [graph classification](https://ogb.stanford.edu/docs/graphprop/) typically involve large numbers of small graphs rather than large individual graphs. In such cases, sampling is not the primary bottleneck during training. Therefore, YOSO and other sampling-based methods do not evaluate on these tasks.
>
> Consequently, we focused our evaluation on node classification and link prediction, where sampling overhead is most impactful. That said, it is evident that improving sampling efficiency would also lead to overall efficiency gains for graph classification tasks, even though they were not our primary focus.
>
> ### 2. Performance metrics
> We agree that the F1-score is an important metric and have used it for evaluation on the Reddit dataset. However, it is not applied across all datasets in our work. This is because, apart from the Reddit dataset, all other datasets are sourced from the standard OGB graph learning benchmarks, which come with their own official evaluation metrics. For example, the OGB benchmarks recommend using Accuracy for ogbn-products and Hits@100 for ogbl-ppa. These metrics are widely adopted in related research. To ensure fair and consistent comparisons, we followed the official recommendations and applied the appropriate metrics for each dataset.
>
> For details on the metrics used for each dataset, please refer to Table 2 in the Appendix A.2.
>
> ## Questions
> ### 1. Potential information loss
> Thank you for this question. In fact, there is no information loss with our method. YOSO trains directly on the original dataset. It does not make the dataset itself sparse but instead transforms the dataset into a specific domain where compression/sampling is performed. At the same time, YOSO can reconstruct the original dataset either losslessly or nearly losslessly, ensuring no (or nearly no) information loss.
>
> ### 2. Data bias
> Metrics such as F1-score are important for the data bias. We used this metric for the Reddit dataset (as it is the official recommendation [1]). Similarly, for other datasets (OGB dataset [2]), we followed the official recommendations and applied the appropriate metrics for each dataset.
> A more detailed explansion can be found in our response to **Weakness 2**.
>
> ### 3. The number of evaluated tasks
> Yes, our method focuses on improving the sampling efficiency of GNN training, which will undoubtedly provide efficiency benefits across all GNN-related tasks.
> The reasons why we only evaluated and reported two graph learning tasks in the paper can be found in our response to **Weakness 1**.
>
>
> **We hope our responses have effectively addressed your concerns. If so, we kindly request that you consider updating your score. Should you have any additional questions or feedback, please don’t hesitate to reach out. We’d be more than happy to provide further clarification.**
>
>
>
>
>
> #### Reference
> [1] Hamilton, Will, Zhitao Ying, and Jure Leskovec. "Inductive representation learning on large graphs." Advances in neural information processing systems 30 (2017).
> [2] Hu, Weihua, et al. "Open graph benchmark: Datasets for machine learning on graphs." Advances in neural information processing systems 33 (2020): 22118-22133.

---

> ### Author Response · Authors · 2024-11-25
> **Follow-Up: Have Our Responses Addressed Your Concerns?**
>
> Dear Reviewer 9rsa,
>
> Thank you again for your valuable feedback and thoughtful suggestions.
>
> With the discussion phase nearing its conclusion (**just 2 days remaining**), we would like to confirm whether our responses have effectively addressed your concerns. If there are any additional questions or areas where further clarification is needed, please do not hesitate to let us know—we would be more than happy to provide additional details.
>
> We sincerely appreciate the time and effort you have invested in reviewing our work. If our revisions and responses have resolved your concerns, we kindly hope you might consider raising your score.
>
> Thank you for your attention and consideration.

---

### Official Review · Reviewer_ZwAn · 2024-11-05

**Soundness:** 2
**Presentation:** 3
**Contribution:** 2
**Rating:** 6
**Confidence:** 3

**Summary:**

This paper introduces YOSO, a novel compressed sensing-based sampling approach for Graph Neural Networks (GNNs) that samples nodes only once per training. By performing a one-time sampling at the input layer and a lossless reconstruction at the output layer, YOSO aims to minimize the computational burden associated with repeated sampling in GNN training. The proposed method is evaluated on node classification and link prediction tasks, demonstrating significant training time reduction (up to 75%) while maintaining comparable accuracy to several baselines.

**Strengths:**

1.	The paper is well-organized and easy to follow. It provides a thorough background on GNNs and compressed sensing, equipping readers with the necessary foundational knowledge. Additionally, the authors offer a detailed discussion on the use of compressed sensing for sampling and the proposed YOSO algorithm.
2.	The compressed sensing-based sampling method is both novel and effective in reducing computational overhead associated with sampling.
3.	The authors present experiments demonstrating YOSO's efficiency on large-scale graph datasets across multiple tasks, supporting the practical impact of their approach.

**Weaknesses:**

1.	The motivation for using compressed sensing-based sampling is unclear. Graph distillation or condensation methods could also be feasible for generating smaller graph datasets while preserving data distribution. What specific advantages does compressed sensing sampling offer over graph condensation?
2.	Compressed sensing relies on the Restricted Isometry Property (RIP) for accurate reconstruction. In the context of graph neural networks, it is unclear if RIP holds for the input feature matrix, representation matrix, and output matrix. Preliminary experiments would be beneficial to validate RIP applicability within GNNs.
3.	This paper proposes using an unknown $U^{(l)}$ and an universal $\Phi$ to enhance the efficiency. However, a key question remains: does the universal $\Phi$ really exist? More justification is needed. Additionally, what’s the disadvantage compared with layer-wise $\Phi$? Is the accuracy loss significant?
4.	For Line 257, I am confused on the forward propagation. Suppose the shape of $\Phi$ is M*N, the normalized Laplacian matrix is N*N, trainable parameters W is d*d, $T^{(l-1)}$ is M*d. It seems that the forward equation is incorrect with an unmatched shape. Is it like $\Phi A \Phi^\top$ to transform the adjacency matrix?
5.	The limitation discussion of YOSO is missing. (1) is YOSO robust over the initialization of sampling matrix $\Phi$? What’s the performance on a random initialized measurement matrix? Why do the authors design a handcraft sampling matrix? (2) YOSO requires more memory during GNN training? What’s the memory consumption of YOSO compared with other methods?

I am open to revising my score if the authors can address my concerns.

---------------------------------------Post Rebuttal----------------------------------

After reading the response, I increase the score to 6.

**Questions:**

Please see the weakness part.

---

> ### Author Response · Authors · 2024-11-21
> **First part of response [Weakness 1]**
>
> Dear Reviewer ZwAn,
> Thank you for your valuable comments. Please find our detailed response blow.
>
> ## Weakness
> ### 1. YOSO V.S. Graph Condensation&Distillation
> Graph Condensation [5] and Graph Distillation [10] are methods designed to enhance computational efficiency. They achieve this by shrinking large-scale graphs into smaller ones while preserving essential structural and feature information. Alternatively, they replace complex GNN models with approximate and computationally simpler models, such as MLPs. However, these kinds of processes introduce additional computational overhead and may result in the loss of important information, potentially leading to a decrease in model performance.
>
> According to the properties of graph condenstation/distillation, YOSO have the following advantages:
> - $\textbf{High accuracy:}$ YOSO has better model accuracy compared to Graph Condensation\&Distillation since YOSO has the ability to reconstruct node embeddings, achieving results equivalent to full-batch GNNs, which have already been proven to be zero-bias and zero-variance [4].
> - $\textbf{Short pre-processing time:}$ Graph Condensation\&Distillation introduces higher pre-processing overhead because, as mentioned earlier, they either rely on expensive computations to obtain a small graph or learn a small, approximate model from a larger one.
>
> To better demonstrate the aforementioned advantages, we conducted new experiments comparing YOSO with two classic graph condensation schemes, GCond [11] and GC-SNTK [12], on the ogbn-arxiv dataset. Both GCond and GC-SNTK use a graph reduction rate of 0.25% on the ogbn-arxiv dataset and paired with the GCN [13]. We evaluated the preprocessing time and the model accuracy for the node classification task. The results are shown below:
>
> |              Dataset              |           ogbn-arxiv           |
> |:---------------------------------:|:------------------------------:|
> |              Schemes              |    GCond \| GC-SNTK \| YOSO    |
> |    Preprocessing Time (second)    | 20615.6 \| 11066.89 \| 1643.32 |
> | Model Accuracy (Metric: Accuracy) |   0.6172 \| 0.6219 \| 0.7169   |
>
> According to the new results, YOSO can achieve higher accuracy and much lower preprocessing time (12X faster than GCond and 6X faster than GC-SNTK.) comapred to the graph condensation-based scheme.
>
> Thank you for bringing these related works to our attention, we have revised the related work section and added.
>
> #### Reference
> [4] Zou, Difan, et al. "Layer-dependent importance sampling for training deep and large graph convolutional networks." Advances in neural information processing systems 32 (2019).
> [5] Gao, Xinyi, et al. "Graph condensation: A survey." arXiv preprint arXiv:2401.11720 (2024).
> [10] Tian, Yijun, et al. "Knowledge distillation on graphs: A survey." arXiv preprint arXiv:2302.00219 (2023).
> [11] Jin, Wei, et al. "Graph Condensation for Graph Neural Networks." International Conference on Learning Representations.
> [12] Wang, Lin, et al. "Fast graph condensation with structure-based neural tangent kernel." Proceedings of the ACM on Web Conference 2024. 2024.
> [13] Kipf, Thomas N., and Max Welling. "Semi-supervised classification with graph convolutional networks." arXiv preprint arXiv:1609.02907 (2016).

---

> ### Author Response · Authors · 2024-11-21
> **Second part of response [Weakness 2]**
>
> ## Weakness
> ### 2. RIP in YOSO design
> Yes, RIP still holds in the context of GNN. Original CS/RIP framework is defined in vector-format [3] and YOSO extends original framework from vector-format to matrix-format. YOSO is equivalent to the original CS/RIP framework, as we can always vectorizing the matrices into vectors. YOSO replaces the vector with matrices (input, representation, and output matrix), uses $|| \cdot ||_{2,1}$ to describe sparsity, and the Frobenius norm to measure the distance.
> **Input feature matrix.** Yes, RIP holds with probability $p=1$. The reason is that the input matrix satisfies all the requirements of CS/RIP (e.g., sparsifiability) and has not participate in GNN computations (e.g., non-linear).
> **Representation and output matrix.** RIP still holds but with a probability $p>1-e^{-cM}$ where $c$ is a constant and $M$ is the number of samples. This statement is proven in Appendix B.3. Although the RIP cannot be fully satisfied ($p=1$), adjusting the sample size $M$ can make the probability of satisfying RIP approach 1. In Section 6.4 of the paper, under 'Reconstruction Effectiveness', we provide a heatmap (Figure 5) to describe the difference between the output matrix reconstructed by YOSO and the output matrix obtained through full-batch (zero-bias, zero-variance) computation. From the results, it can be observed that when $M$ increases to a certain value (e.g., $M=512$ in Figure 5), the reconstruction error becomes negligible.
>
> Thank you for pointing this out. We've add these clarification into our revised manuscript and the changes are highlighted in blue.
>
>
> #### Reference
> [3] Davenport, Mark A., et al. "Introduction to compressed sensing." (2012): 1-64.

---

> ### Author Response · Authors · 2024-11-21
> **Third part of response [Weakness 3]**
>
> ## Weakness
> ### 3. Existence of $\mathbf{\Phi}$ and disadvantage of layer-wise $\mathbf{\Phi}$
> **The Existence of universal $\mathbf{\Phi}$.** Yes, universal $\mathbf{\Phi}$ exists. In the early work of CS, to expand its application scope, domain-specific sampling matrices were designed for data types where the sparse domain was known. For example, in image data, the Fourier transform domain is explicitly sparse [6], allowing for targeted design of the sampling matrix. However, in many fields, the specific sparse domain is unknown, presenting a challenge similar to YOSO: assuming the data can be sparsified, does a $\mathbf{\Phi}$ exist without violating RIP? In [1], a construction method for a sampling matrix (random mapping) is provided, demonstrating that a universal sampling matrix does indeed exist. In other words, [1] has already provided a proof of the existence of a universal sampling matrix.
>
> **How do we find the universal $\mathbf{\Phi}$ in the context of GNN?** In [1], the existence proof is provided along with another conclusion: although $\mathbf{\Phi}$ exists, the sampling matrix $\mathbf{\Phi}$ constructed using different methods will require different numbers of rows $M$ (i.e., different sample sizes) to satisfy the RIP. In the YOSO design, our starting point for selecting a specific $\mathbf{\Phi}$ is to minimize $M$ as much as possible without violating the RIP. Firstly, to minimize $M$, for any given graph data, we need to retain its 'most important' $M$ nodes. This importance is measured in graph signal processing [7,8,9] by the top $M$ largest eigenvalues of a matrix that reflects certain graph structures, such as the adjacency matrix or the normalized Laplacian (As we demonstrated in the 'Construction of $\hat{\mathbf{S}}$' of Section 5.2.). Secondly, as we demonstrated in the 'Construction of $\mathbf{\Sigma}$' of Section 5.2, the role of $\mathbf{\Sigma}$ comes from two aspects: first, according to [1], it needs to maintain the necessary randomness; second, it indicates the contribution level of each node $i$ to the non-zero rows (which corresponds to the concept of support [2,3] in CS). Finally, by computing $\mathbf{S} \otimes \mathbf{\Sigma}$, we can obtain the universal sampling matrix $\mathbf{\Phi}$.
>
> **Layer-wise $\mathbf{\Phi}$ V.S. universal $\mathbf{\Phi}$.** A notable drawback of using layer-wise $\mathbf{\Phi}$ is that its computational cost is disproportionately high compared to the improvement it brings in model accuracy. Specifically, to make this statement more clearly, we conducted the following experiments: on the datasets ogbn-arxiv and ogbl-ppa, we compared the total training time and model accuracy when using layer-wise $\mathbf{\Phi}$ and universal $\mathbf{\Phi}$. The results are shown in the table below.
>
> |       Dataset       |        ogbn-arxiv       |         ogbl-ppa        |
> |:-------------------:|:-----------------------:|:-----------------------:|
> |         Type        | Layer-wise \| Universal | Layer-wise \| Universal |
> | Total Training Time (second)|      59.22 \| 10.93     |      145.5 \| 21.46     |
> |    Model Accuracy   |      0.73 \| 0.727      |     0.2254 \| 0.2235    |
>
> As seen in the table, across two different learning tasks, the layer-wise based scheme increases the total training by 5X (ogbn-arxiv) and 7X (ogbl-ppa), yielding only a marginal accuracy improvement at the 0.001 level compared to the universal-based scheme. Therefore, using a universal $\mathbf{\Phi}$ does not lead to a significant reduction in accuracy.
>
>
> #### Reference
> [1] Candes, Emmanuel J., and Terence Tao. "Near-optimal signal recovery from random projections: Universal encoding strategies?." IEEE transactions on information theory 52.12 (2006): 5406-5425.
> [2] Vaswani, Namrata, and Wei Lu. "Modified-CS: Modifying compressive sensing for problems with partially known support." IEEE Transactions on Signal Processing 58.9 (2010): 4595-4607.
> [3] Davenport, Mark A., et al. "Introduction to compressed sensing." (2012): 1-64.
> [6] Gilbert, Anna C., et al. "Recent developments in the sparse Fourier transform: A compressed Fourier transform for big data." IEEE Signal Processing Magazine 31.5 (2014): 91-100.
> [7] Ortega, Antonio, et al. "Graph signal processing: Overview, challenges, and applications." Proceedings of the IEEE 106.5 (2018): 808-828.
> [8] Leus, Geert, et al. "Graph Signal Processing: History, development, impact, and outlook." IEEE Signal Processing Magazine 40.4 (2023): 49-60.
> [9] Dong, Xiaowen, et al. "Graph signal processing for machine learning: A review and new perspectives." IEEE Signal processing magazine 37.6 (2020): 117-127.

---

> ### Author Response · Authors · 2024-11-21
> **Fourth part of response [Weakness 4, 5]**
>
> ## Weakness
> ### 4. The shape of matrics in forward propagation
> Thank you for pointing out this issue of inconsistent mathematical notation. Yes, we also transform the normalized Laplacian matrix $\hat{\mathbf{A}}$ by using sampling matrix $\mathbf{\Phi}$ through $\mathbf{\Phi}\hat{\mathbf{A}}\mathbf{\Phi}^{T}$.
> We have revised this in the updated paper and highlighted the change with blue.
>
> ### 5. The limitation of YOSO
> **Robustness.** Yes, YOSO's $\mathbf{\Phi}$ is not fixed, with part of it derived from a sub-Gaussian distribution. However, regardless of the specific values, this $\mathbf{\Phi}$ can robustly function without violating the RIP.
> **Handcraft $\mathbf{\Phi}$.** Our motivation for manually designing $\mathbf{\Phi}$ is to ensure that YOSO satisfies the RIP. Since RIP determines whether embeddings can be perfectly reconstructed, it directly impacts the final model accuracy of YOSO.
> **The influence of the random initiliazion.** If a purely random $\mathbf{\Phi}$ is used, according to [7, 8], it has high chance of failing to satisfy the RIP, resulting in poor model accuracy.
> **Memory consumption.** No, YOSO does not require more memory during GNN training. We added an experiment on the ogbn-arxiv dataset, selecting one baseline from each sampling scheme (node-wise[GraphSage], layer-wise[FastGCN], subgraph-based[ClusterGCN]) to test GPU memory usage. The results are shown in the table below.
>
> |         Dataset        |                 ogbn-arxiv                 |
> |:----------------------:|:------------------------------------------:|
> |         Schemes        | GraphSage \| FastGCN \| ClusterGCN \| YOSO |
> | GPU Memory Consumption |       451.46 \| 4.58 \| 131.46 \| 3.1      |
>
> It can be observed that YOSO has the smallest GPU consumption, which benefits from its two key features: (1) YOSO can maintain a much small sample size without sacrificing accuracy (due to the ability to reconstruct); and (2) it samples only once and uses the sampled data throughout the entire training process.
>
> **We hope our responses have effectively addressed your concerns. If so, we kindly request that you consider updating your score. Should you have any additional questions or feedback, please don’t hesitate to reach out. We’d be more than happy to provide further clarification.**
>
>
> #### Reference
> [7] Ortega, Antonio, et al. "Graph signal processing: Overview, challenges, and applications." Proceedings of the IEEE 106.5 (2018): 808-828.
> [8] Leus, Geert, et al. "Graph Signal Processing: History, development, impact, and outlook." IEEE Signal Processing Magazine 40.4 (2023): 49-60.

---

> ### Author Response · Authors · 2024-11-25
> **Follow-Up: Have Our Responses Addressed Your Concerns?**
>
> Dear Reviewer ZwAn,
>
> Thank you once again for your constructive feedback and insightful suggestions.
>
> As we approach the end of the discussion phase (**only 2 days left**), we would like to check if our responses have sufficiently addressed your concerns. Should there be any additional questions or points requiring further clarification, please feel free to let us know, we would be more than happy to provide further details.
>
> We greatly appreciate the time and effort you have dedicated to reviewing our submission. If our revisions and responses have resolved your concerns, we kindly hope you will consider raising your score.
>
> Thank you for your time and thoughtful consideration.

---

> > ### Comment · Reviewer_ZwAn · 2024-11-25
> >
> > Thank you for your response and the improvements made to the paper! I will follow the ongoing discussions with the other reviewers and finalize my decision afterward.

---

> > > ### Author Response · Authors · 2024-12-01
> > > **Follow-up on Review Feedback**
> > >
> > > Dear Reviewer ZwAn,
> > >
> > > Thank you for acknowledging our efforts in addressing your comments in the rebuttal. We greatly appreciate the time and effort you’ve devoted to reviewing our work.
> > >
> > > As the discussion phase is nearing its conclusion, we wanted to kindly check if you have any additional concerns or suggestions regarding our revisions. If there are any remaining points, we would be happy to address them promptly. If you believe our updated paper meets your expectations, we hope you might consider reflecting that in your evaluation.
> > >
> > > Thank you once again for your valuable feedback.

---

### Official Review · Reviewer_Yg8e · 2024-11-06

**Soundness:** 3
**Presentation:** 3
**Contribution:** 3
**Rating:** 5
**Confidence:** 5

**Summary:**

Paper is interested in sampling graphs, for the purpose of training graph neural networks, in cases where the input graph is large. Rather than taking gradient-steps using entire graphs, sampling-based training of GNNs can take gradient steps using subgraphs (more compute efficient).

Paper shows while indeed existing graph-sampling methods *can* scale learning onto larger graphs, *however*, they significant time in sampling itself (i.e., data prep) rather than on gradient calculation (i.e., training).

**Strengths:**

* GNNs are indeed popular architectures, solving problems across many domains. Even though there are many methods are proposes to speed-up training of GNNs, making things even faster, should realize some advantages.

* Their sampling method takes less time than competition, yet reaches SOTA metrics.

* Their sampling is computed only once, whereas competing methods draw subgraph samples for every input example every epoch. Since sampling time is significant (often the dominant), then sampling only once significantly saves resources.

**Weaknesses:**

# Related Work

* Paper does not address non-sampling-based methods for speeding-up GNNs. At least, they should be mentioned in related work (IMO). One missed family is historical embeddings (e.g., https://arxiv.org/abs/1710.10568, https://arxiv.org/abs/2106.05609, https://arxiv.org/abs/2305.12322), another family is some "linearization" of models (a.k.a "decoupled" GNNs), e.g., https://arxiv.org/pdf/1902.07153, https://arxiv.org/abs/2004.11198, https://arxiv.org/abs/2111.06312)

* Other methods also sample-once. E.g., ClusterGCN.

# Sampling is inherently parallelizable

Paper is founded upon a statement that sampling takes more time than the actual training step. I believe the statement is correct (because the graph may be large and does not fit in memory). However, sampling is trivially distributable -- in fact, many papers, such as, PinSAGE, TFGNN, ..., propose and implement distributed or multi-threaded sampling. I think this could also be mentioned in the related work.


# Possible Writing Improvements

* First line of Intro: "analyzing" should probably become "modeling"
* The abstract says "lossless" compression but line 90 says "nearly lossless". It is better to be consistent.
* The term "embedding matrix" is used a few times in the first 2 pages, without clear definition of what it is -- is it the node input features? is it the hidden representations (between layers)? Is it the output of the GNN? I feel that it is all of those [but only after reading a couple of more pages].


# Missing analysis

The paper talks about bias and variance. This is usually done **either** in the forward pass -- let $\widetilde{z}$ be the latent values obtain via the graph sample and $\mathbf{z}$ be the latent values when full graph is used, the zero-biased analysis should show $\mathbb{E}[\widetilde{z}] = \mathbf{z}$; or in the backward pass e.g. $\mathbb{E}[\frac{\partial loss(\widetilde{z})}{\partial \theta}] = \frac{\partial loss(\mathbf{z})}{\partial \theta}$. This paper does neither of those, nor it shows bounds on the variance of these quantities. I suggest the authors remove bias&variance arguments or else show some analysis to back the arguments.

# Math inaccuracies
* Adjacency matrix is square (N x N). How is it multiplied on the right by the model parameter matrix? Are GNN parameters a function of the input graph? This happens in Eq6 and Lines 5&7 in Alg1.

* Line 1 in Alg1 does not explain the initialization process. Crucially, is $U$ initialized to be a(n orthonormal) basis?

* line 161 "exists" must be a function of k and/or the rank of $\widehat{H}$ -- for example, what if I choose $k=0$, then there is no orthonormal basis $U$ that can recover $H$. In fact, $rank(\widehat{H})$ must be at least equal the rank of $H$ for recovery to be possible.

* The sampling matrix (described below Eq1) can be better described. Can you explain its structure? or its motive? Even a google search on "compressed sensing sampling matrix" does not pull-up the answer immediately. What is its rank?

* Most-crucially, Eq2, which the rest of the work is founded upon, reads wrong/incomplete. Most-likely, *there are trivial typos, however, I will not do deductions and I expect all authors to ensure correctness of their **main** equation*. $\widetilde{H}$ reads as a scalar (the minimum norm).

* Line 339: eigenvalues do not correspond to nodes. They correspond to eigenvectors. For instance, one can do a low-rank representation of the adjacency matrix (effectively using few eigenvalues).

**Questions:**

+ Does the (sparse reconstruction) optimization problem have to be solved at every layer, for every new graph? Or is the assumption that the graph is always fixed (between training and inference)?

+ If the parameters of the model significantly change, isn't the re-sampling necessary? Why not sample with some periodicity (as model drifts), instead of sample only once at-start?

+ Is it at-all possible to think about the GNN in the inductive setting in your framework?

---

> ### Author Response · Authors · 2024-11-21
> **First part of the response [Weakness 1]**
>
> Dear Reviewer Yg8e,
> Thank you for your valuable comments. Please find our detailed response blow.
>
> ## Weakness
> ### 1. Related Work
> Thank you for bringing these related works to our attention, including non-sampling-based methods and linearization model approaches. We have addressed these methods and outlined their connections to our work as follows:
> **Historical Embedding.** This class of methods is not independent of sampling. Instead, they are often integrated with existing sampling strategies to improve specific aspects of sampling performance, such as estimated variance [1], or expressiveness [2]. For example, the first reference you mentioned, [VR-GCN](https://arxiv.org/abs/1710.10568), which we have already included as a baseline in our experiments, utilizes historical embeddings within node-wise sampling. The second reference, [GNNAutoScale](https://arxiv.org/abs/2106.05609), incorporates the concept of historical embeddings within subgraph-based sampling. Although historical embedding can be effective in terms of accuracy, they often come with high computational complexity. Figure 2(a)-(e) includes two node-wise sampling methods: VR-GCN and GraphSage. Compared to GraphSage, the sampling time for VR-GCN increased by 23.68% on the ogbn-products dataset and by 21.95% on the ogbl-citation2 dataset. This overhead is caused by VR-GCN's integration of historical embeddings, which further intensifies the computational burden.
> **Linearization.** Following your suggestion, we have added experiments to evaluate the performance of the baselines you mentioned, including [SIGN](https://arxiv.org/abs/2004.11198) and [iSVD](https://arxiv.org/abs/2111.06312) on ogbn-products and ogbn-arxiv dataset, respectively. We use iSVD-best to represent the version of this baseline with the highest accuracy, specifically $iSVD_{250}+dropout(LR)+dropout(\hat{M}_{LR}^{(NC)})+finetune  \mathbf{H}$, as mentioned in the paper. The results are presented in the table below.
>
> | Dataset | ogbn-products | ogbn-arxiv |
> |:---:|:---:|:---:|
> | Schemes | SIGN-2 \| SIGN-4 \| SIGN-6 \| SIGN-8 \| YOSO | iSVD \| iSVD-best \| YOSO |
> | Total Training Time | 421.79 \| 584.07 \| 831.94\| 1052.96 \| 499.02 | 9.94 \| 982.12 \| 10.74 |
> | Model Accuracy | 0.761 \| 0.778 \| 0.776 \| 0.783 \| 0.788 (Metric: Accuracy) | 0.685 \| 0.746 \| 0.72 (Metric: Accuracy) |
>
> From the results in the table, it can be observed that for SIGN, its accuracy on the ogbn-products dataset does not exceed that of YOSO (0.788). While SIGN-2 achieves a total training time that is 18.3% lower than YOSO, its accuracy drops by 2.7%. A similar trend is observed for the iSVD baseline. The low-accuracy version of iSVD reduces total training time by 8% but suffers an accuracy drop of 4%. In contrast, the high-accuracy version, iSVD-best, increases total training time by 91X compared to YOSO, with only a 0.014 improvement in accuracy.
>
> **ClusterGCN.** ClusterGCN is a representative of subgraph-based sampling, and is already listed it as a baseline in our paper. While it adopts a sample-once strategy, its sampling process encounters the computational challenges of the NP-Hard or NP-Complete graph partition problem. In contrast, our proposed method circumvents this issue by not relying on such partitioning, thereby providing a more efficient and scalable alternative. As shown in Figure 2 and Table 1 of our paper, ClusterGCN, as an example, compared to YOSO, increases the total training time by 12X on ogbn-arxiv (node classification) and 9X on ogbl-ppa (link prediction). However, the model accuracy decreases by 10% and 9%, respectively.
>
> Thank you again for your suggestions. We have revised the related work section and added all the above discussion in our updated paper. The changes are highlighted in blue.
>
>
> #### Reference
> [1] Chen, Jianfei, Jun Zhu, and Le Song. "Stochastic Training of Graph Convolutional Networks with Variance Reduction." International Conference on Machine Learning. PMLR, 2018.
> [2] Fey, Matthias, et al. "Gnnautoscale: Scalable and expressive graph neural networks via historical embeddings." International conference on machine learning. PMLR, 2021.

---

> ### Author Response · Authors · 2024-11-21
> **Second part of the response [Weakness 2, 3, 4]**
>
> ## Weakness
> ### 2. Sampling is inherently parallelizable
> **Parallel/Distributed Computing is orthogonal to YOSO.** Optimization techniques for GNN sampling have been well explored at the algorithmic level, system level, and hardware level, as detailed in [3,4]. These approaches are orthogonal to one another. For instance, at the hardware level, specialized accelerators can be developed to improve the efficiency of sampling. At the system level, techniques such as multi-threading and pipelining can be employed to enhance computational performance. YOSO focuses on algorithm-level optimization. These orthogonal approaches can complement each other, meaning YOSO could benefit from system-level or hardware-level enhancements while maintaining its core algorithmic improvements.
>
> We've added and highlighted this discussion in Related Work.
>
> ### 3. Possible Writing Improvements
> We sincerely appreciate your thoughtful and valuable feedback on our manuscript. We have incorporated the writing improvements you suggested in the paper and highlighted them in blue.
>
> To clarify the inconsistency concern regarding \'lossless\' and \'nearly lossless\': In the original YOSO design, to ensure accuracy, each layer of the GNN performs a sampling-reconstruction operation. While this guarantees losslessness, it incurs significant costs, as discussed in the two challenges outlined in Section 4 of the paper. While this guarantees losslessness (meaning it satisfies the RIP condition with probability $p=1$), it incurs significant costs, as discussed in the two challenges outlined in Section 4 of the paper. Therefore, we trade some degree of losslessness for faster computation time, making YOSO nearly lossless. The extent of this \'nearly\' is theoretically tightly estimated in Appendix B.2, where we prove that under this trade-off condition, the RIP is satisfied with probability $p$, and $p > 1-e^{-cM}$, where $c$ is a positive constant and $M$ is the number of samples.
>
> We appreciate your feedback and have carefully revised our paper to clarify this point and prevent any further confusion.
>
> ### 4. Missing analysis
> We appreciate your suggestion and agree that bias and bias-variance analysis are typically important for GNN sampling methods. However, we firmly believe that such an analysis is not necessary for our method, YOSO, for the following reasons:
> - **YOSO achieves results equivalent to full-batch.** Based on compressed sensing, which ensures that the original data can be perfectly reconstructed from a small number of samples, YOSO is able to compute using only a few sampled nodes, which are then reconstructed back to the original data, as demonstrated in Equation(1)-(5) in the Section3 and Section4 of the paper. This means that although YOSO only utilizes a subset of the data, it can achieve results equivalent to full-batch (using all data).
> - **Full-batch is zero-bias\&zero-variance.** In [5], the authors point out that full-batch (non-sampling) GNNs (such as GCN [6]) yield zero-bias/zero-variance, with detailed mathematical derivations provided. Therefore, YOSO is also zero-bias\&zero-variance.
>
> To clarify this logic as well as the concepts of bias and variance, we have revised parts of the introduction to make it more understandable, highlighting the changes in blue.
>
>
> #### Reference
> [3] Zhang, Shichang, et al. "A survey on graph neural network acceleration: Algorithms, systems, and customized hardware." arXiv preprint arXiv:2306.14052 (2023).
> [4] Abadal, Sergi, et al. "Computing graph neural networks: A survey from algorithms to accelerators." ACM Computing Surveys (CSUR) 54.9 (2021): 1-38.
> [5] Zou, Difan, et al. "Layer-dependent importance sampling for training deep and large graph convolutional networks." Advances in neural information processing systems 32 (2019).
> [6] Kipf, Thomas N., and Max Welling. "Semi-supervised classification with graph convolutional networks." arXiv preprint arXiv:1609.02907 (2016).

---

> ### Author Response · Authors · 2024-11-21
> **Third part of the response [Weakness 5]**
>
> ## Weakness
> ### 5. Math inaccuracies
> **Normalized Laplacian matrix ($\hat{\mathbf{A}}$) multiplication.** Thank you for pointing out this issue with inconsistent mathematical notation. Sorry for the confusion caused. In fact, we also transform the normalized Laplacian matrix $\hat{\mathbf{A}}$ by applying the sampling matrix $\mathbf{\Phi}$ as $\mathbf{\Phi}\hat{\mathbf{A}}\mathbf{\Phi}^{T}$. We have revised this in the updated paper and highlighted the change in blue.
> **Initialization process in the pseudocode.** $\mathbf{U}$ is not initialized as orthogonal but is iteratively adjusted to become orthogonal during the training process (Line 16 in Algorithm 1 of the paper). We have revised the description regarding the initialization in the updated paper and highlighted the change in blue.
> **'Exists' statement and value of $k$.** No, 'exists' is not a function of $k$ or $\hat{\mathbf{H}}$. In the field of graph signal processing, the existence of $\mathbf{U}$ has been mathematically proven [7,8]. Its existence does not depend on the sparsity level $k$. Moreover, the sparsity level $k$ cannot be manually specified; it is not a hyperparameter but is determined by the graph itself [9, 10].
> **Sampling matrix $\mathbf{\Phi}$.** In Section 5.2 of the paper, we provide a detailed description of the structure, motivation, and construction method of the sampling matrix. Its structure is an $M \times N$ (where $M \ll N$) matrix, and the primary design motivation is to ensure that $\mathbf{\Phi}$ can work with ($\mathbf{\Phi U}$) any $\mathbf{U}$ without violating the RIP (as described in Equation (3) of the paper). We refer to it as the sampling matrix because, in the original CS theory, a matrix that performs a similar task is called a sensing matrix. The sensing matrix's main role is to compress certain raw data, and if the sensing matrix is known, the compression process is reversible. In the context of GNNs, YOSO uses $\mathbf{\Phi}$ not only to perform compression but also to enable further computation on the compressed data. Moreover, $\mathbf{\Phi}$ serves the function of a sampling operator (as represented by the $\mathcal{P}$ function in Section 3 of the paper). Therefore, we refer to it as the sampling matrix.
> **Equation(2) of the paper.** We would like to clarify: when dealing with the optimization problem in CS, we face an underdetermined linear system (Equation (2) of the paper), where the number of unknowns exceeds the number of equations, resulting in infinitely many solutions. Among these solutions, we are interested in the sparsest one. Therefore, our original intention was that $\tilde{\mathbf{H}}$ refers to the sparsest solution, rather than a scalar derived from the $l_{2,1}$-norm. We have revised this in the updated paper and highlighted the change in blue.
> **Eigenvalues and nodes.** In graph signal processing, associating eigenvalues with nodes is a common approach [7,8,9,10]. For example, if the node IDs are predefined, a fixed matrix reflecting the graph structure (such as the adjacency matrix or the normalized Laplacian) can be obtained. By performing eigenvalue decomposition on this matrix, the eigenvalues and eigenvectors corresponding to each node ID are determined, thus creating an association between eigenvalues and nodes. The advantage of this approach lies in the fact that the eigenvalues and eigenvectors corresponding to a particular node capture its relative importance or influence within the entire graph. This method is also widely used in spectral graph theory [11].
>
>
> #### Reference
> [7] Ortega, Antonio, et al. "Graph signal processing: Overview, challenges, and applications." Proceedings of the IEEE 106.5 (2018): 808-828.
> [8] Leus, Geert, et al. "Graph Signal Processing: History, development, impact, and outlook." IEEE Signal Processing Magazine 40.4 (2023): 49-60.
> [9] Anis, Aamir, Akshay Gadde, and Antonio Ortega. "Efficient sampling set selection for bandlimited graph signals using graph spectral proxies." IEEE Transactions on Signal Processing 64.14 (2016): 3775-3789.
> [10] Hu, Wei, et al. "Graph signal processing for geometric data and beyond: Theory and applications." IEEE Transactions on Multimedia 24 (2021): 3961-3977.
> [11] Spielman, Daniel. "Spectral graph theory." Combinatorial scientific computing 18 (2012): 18.

---

> ### Author Response · Authors · 2024-11-21
> **Fourth part of response [Questions 1, 2, 3]**
>
> ## Questions
> ### 1. Reconstruction problem of every layer and the new graph
> - **Every layer.** No. Reconstruction does not need to be performed at every layer. Performing reconstruction at every layer would incur a very high computational cost while yielding minimal benefits in terms of reconstruction quality or error (which is related to the final model accuracy). The reason for this is proven in Appendix B.3 of the paper: even if reconstruction is only performed at the final layer, the RIP (which determines whether perfect reconstruction is possible) can still be satisfied with probability $p$ where $p \geq (1-e^{-cM})$ ($c>0$ is a constant.).
> - **New Graph.** Yes, reconstruction is necessary. However, this is not unique to YOSO. For most sampling-based GNN models, such as FastGCN, AS-GCN, and GraphSAINT, a new graph implies a change in the graph structure, which requires re-training before inference can be performed.
>
> ### 2. Parameters of the model significantly change
> No, re-sampling is not necessary. YOSO's theory is independent of the magnitude of parameter changes during training. Sampling with some periodicity would only increase the overhead of sampling time but would not impact the final results. The reason is that, YOSO's goal is to use embeddings to reconstruct the original embeddings (equivalent to full-batch), and the magnitude of changes does not affect this reconstruction process.
>
> ### 3. Transductive and inductive
> Thank you for this valuable feedback. We acknowledge that YOSO is currently transductive, which aligns with most sampling-based models, such as FastGCN, AS-GCN, LADIES, and GraphSAINT. After careful consideration, it is possible to change current YOSO to be inductive by changing the construction process of the sampling matrix. These changes fall beyond the scope of this work, and we leave them as future work.
>
>
> **We hope our responses have effectively addressed your concerns. If so, we kindly request that you consider updating your score. Should you have any additional questions or feedback, please don’t hesitate to reach out. We’d be more than happy to provide further clarification.**

---

> > ### Comment · Reviewer_Yg8e · 2024-11-22
> >
> > You've done several improvements to your paper! I will upvote by 1 for now. I will also be watching discussions with other reviewers

---

> > > ### Author Response · Authors · 2024-12-01
> > > **Follow-up on Review Feedback**
> > >
> > > Dear Reviewer Yg8e,
> > >
> > > Thank you for acknowledging our efforts in addressing your comments in the rebuttal. We greatly appreciate the time and effort you’ve devoted to reviewing our work.
> > >
> > > As the discussion phase is nearing its conclusion, we wanted to kindly check if you have any additional concerns or suggestions regarding our revisions. If there are any remaining points, we would be happy to address them promptly. If you believe our updated paper meets your expectations, we hope you might consider reflecting that in your evaluation.
> > >
> > > Thank you once again for your valuable feedback.

---

### Meta-Review · Area_Chair_Xwjy · 2024-12-26

**Metareview:**

This paper proposes a new sampling algorithm for efficient GNN training using compressed sensing. Although there has been extensive work in this area, the proposed sampling method is different from previous ones and has the benefit of sampling only once. Although this is an important area and the proposed method seems promising, there’s plenty of clarification issues in the paper, and as reviewers pointed out in the final discussion, even the revision submitted by the authors contain many errors in key formulas and symbols. Therefore, we think the current paper does not meet the ICLR standard.

**Additional Comments On Reviewer Discussion:**

Reviewers raised many issues about the definition, symbols, and equations in the original paper. Although the authors are able to clarify many of them in the rebuttal, reviewers still think that the revision does not meet the publication standard of ICLR.

---

### Decision · Program_Chairs · 2025-01-22

Reject